

# Spatial and temporal variability of rainfall and their effects on hydrological response in urban areas - a review

Elena Cristiano[1], Marie-claire ten Veldhius[1], and Nick van de Giesen[1]

[1]Department of Water Management, Delft University of Technology, Postbox 5048, 2600 GA, Delft, The Netherlands

*Correspondence to:* Elena Cristiano (E.Cristiano@tudelft.nl)

**Abstract.** In urban areas, hydrological processes are characterised by high variability in space and time, making them sensitive to small-scale temporal and spatial rainfall variability. In the last decades new instruments, techniques and methods have been developed to capture rainfall and hydrological processes at high resolution. Weather radars have been introduced to estimate high spatial and temporal rainfall variability. At the same time, new models have been proposed to reproduce hydrological response, based on small-scale representation of urban catchment spatial variability. Despite these efforts, interaction between input variability and model resolution remains poorly understood, and further investigations are needed. This paper presents a review of our current understanding of hydrological processes in urban environments as reported in the literature, focusing on their spatial and temporal variability. We review recent findings on the effects of rainfall variability on hydrological response and identify gaps where knowledge need to be further developed to improve our understanding of and capability to predict urban hydrological response.

## 1 Introduction

The lack of sufficient information about spatial distribution of short-term rainfall can be considered as one of the most important sources of errors in urban runoff estimation (Niemczynowicz, 1988a). In the last years considerable advances in quantitative estimation of distributed rainfall data have been made, thanks to new technologies, in particular weather radars (Leijnse et al., 2007; van de Beek et al., 2010; Otto and Russchenberg, 2011). This is a promising development, especially for urban hydrology research, where the hydrological response is sensitive to small-scale rainfall variability in both space and time (Faures et al., 1995; Emmanuel et al., 2012; Smith et al., 2012; Ochoa-Rodriguez et al., 2015b), due to a typically high degree of imperviousness and to a high spatial variability of urban land use.

Progress in rainfall estimation is accompanied by increasing availability of high resolution topographical data, especially digital terrain models and land-use distribution maps derived (Mayer, 1999; Fonstad et al., 2013). High resolution topographical datasets have promoted development of more detailed and more complex numerical models for predicting flows (Gironás et al., 2010; Smith et al., 2013)). However, model complexity and resolution need to be balanced with the availability and quality of rainfall input data and datasets for catchment representation (Morin et al., 2001; Rafieeinasab et al., 2015; Rico-Ramirez et al., 2015; Rafieeinasab et al., 2015; Pina et al., 2016). This is particularly critical in urban hydrology, where flows are sensitive to variations at small space and time scales as a result of the fast hydrological response and the high catchment





variability (Fabry et al., 1994; Singh, 1997). In urban areas, alterations of natural flows introduced by human interventions, especially artificial drainage networks, sewer pipe networks, detention facilities and control facilities (such as pumps and weirs) complicate prediction of flows. Recently, various authors investigated the sensitivity of spatial and temporal rainfall variability on the hydrological response for urban areas (Bruni et al., 2015; Ochoa-Rodriguez et al., 2015b; Rafieeinasab et al., 2015).
Despite these efforts, many aspects of hydrological processes in urban areas remain poorly understood.

It is timely to review recent progress in understanding of interactions between rainfall spatial and temporal resolution, variability of catchment properties and their representation in hydrological models. Section 2 of this paper is dedicated to definitions of spatial and temporal scales and catchments in hydrology and methods to characterise these. Section 3 focuses on rainfall, analysing the most used rainfall measurement techniques, their capability to accurately measure small-scale spatial and
temporal variability, with particular attention to applications in urban areas. Hydrological processes are described in Section 4, highlighting their variability and characteristics in urban areas. Thereafter, the state of the art of hydrological models, as well as their strengths and limitations to account for spatial and temporal variability, are discussed. Section 6 presents recent approaches to understand the effect of rainfall variability in space and time on hydrological response. Section 7, main knowledge gaps are identified for the with respect to accurate prediction of urban hydrological response in relation to spatial
and temporal variability of rainfall and catchment properties in urban areas.

## 2  Scales in urban hydrology

### 2.1  Spatial and temporal scale definitions

Hydrological processes occur over a wide range of scales in space and time, varying from 1 mm to 10000 km in space and from seconds up to 100 years in time. A scale is defined here as the characteristic region in space or period in time at which
processes take place or the resolution in space or time at which processes are best measured (Salvadore et al., 2015).

Several authors have classified hydrological process scales and variability, focusing in particular on the interaction between rainfall and the other hydrological processes (Blöschl and Sivapalan, 1995; Bergstrom and Graham, 1998). Blöschl and Sivapalan (1995): presented a graphical representation of spatial and temporal variability of the main hydrological processes on a logarithmic plane. The plot has been updated by other authors, each focusing on specific aspects. For example, Salvadore et al.
(2015) analysed phenomena related to urban processes, focusing small spatial scale, while Van Loon (2015), added scales of some hydrological problems, such as flood and drought. Figure 1 presents an updated version of the plot that integrates the information contribyted by Berndtsson and Niemczynowicz (1986), Blöschl and Sivapalan (1995), Stahl and Hisdal (2004) and Salvadore et al. (2015). Figure 1 shows that in urban hydrology attention is mainly focused on small scales. Characteristic processes, such as storm drainage, infiltration and evaporation vary at a small temporal and spatial scale, from seconds to hours
and from centimetres to hundreds of meters. Many processes are driven by rainfall, that varies over a wide range of scales.

Blöschl and Sivapalan (1995) highlighted the importance of making a distinction between two types of scales: the "process scale", i.e. the proper of the considered phenomenon, and the "observation scale", related to the measurement and depending on techniques and instruments being used. In the best scenario, process and observation scale should coincide, but this is





not always the case. In order to match observation with process scale, it is often necessary to scale the measurements. These transformations are based on downscaling and upscaling techniques (Fig. 2), as discussed in section 2.2.

## 2.2 Downscaling and upscaling in hydrology

The term downscaling usually refers to methods used to take information known at large scale and make predictions at small scale. There are two main downscaling approaches: dynamic or physically based and statistical methods (Xu, 1999). Dynamic downscaling approaches solve the process-based physics dynamics of the system. In statistical downscaling, a statistical relationship is defined between local variables and large scale prediction and this relationship is applied to simulate local variables (Xu, 1999). Dynamical downscaling is widely used in climate modelling and numerical weather prediction, while statistical models are often used in meteohydrology, for example rainfall downscaling. Dynamic downscaling models have the advantage to be physically-based, but they require a lot of computational power compared to statistical downscaling models. Statistical approaches require historical data and knowledge of local condition.

Statistical downscaling approaches are reported in the literature for a wide variety of variables (Rummukainen, 1997; Deidda, 2000; Ferraris et al., 2003; Gires et al., 2012; Wang et al., 2015; Muthusamy et al., 2016) and techniques such as regression methods, weather pattern-based approaches and stochastic weather generators (see Wilby and Wigley (1997) for a review). Ferraris et al. (2003) presented a review of three common stochastic downscaling models, mainly used for spatial rainfall downscaling: multifractal cascades, autoregressive processes and point process models based on the presence of individual cells. The first were introduced in the 1970s and are widely used to reproduce the spatial and temporal variability (see Schertzer and Lovejoy (2011) for a review). Autoregressive processes are mainly used to generate multidimensional random fields and they take into account the spatial autocorrelation of the rainfall field. The last model is used when the spatial structure of intense rainfall field is defined by convective cells. It incorporates local information and requires a more detailed storm cell identification.

## 2.3 Methods to characterize hydrological process scales

### 2.3.1 Spatial variability of basin characteristics

Slope, degree of imperviousness, soil properties and many other catchment characteristics are variable in space and time and this variability affects the hydrological response (Singh, 1997). This is especially the case of urban areas, where spatial variability and temporal changes in land-use are typically high.

Julien and Moglen (1990) gave a first definition of the catchment length scale $L_s$ as part of a theoretical framework, where they analysed $8400$ dimensionless hydrographs obtained from one-dimensional finite element models under spatially varied input. Length scale was presented as function of rainfall duration $t_c$, spatially averaged rainfall intensity $i$, average slope $s_0$ and average roughness $n$:

$$L_s = \frac{t_c^{\frac{5}{6}} s_0^{\frac{1}{2}} i^{\frac{2}{3}}}{n} \tag{1}$$



The length scale is derived starting from the Manning equation, assuming rainfall duration $t_c$ equal to the time of equilibrium $t_e$, and it depends on both catchment and storm characteristics. For a field with runoff length shorter than $L_s$, the spatial variability has a small influence on the runoff generation, while for runoff length longer than $L_s$, the runoff discharge is strongly affected. The length scale can be useful to determine critical hydrological model grid size: a long duration and low intensity rainfall event, for example, can be analysed using coarse resolution models (Julien and Moglen, 1990).

### 2.3.2 Time scale characteristics

Hydrological response time parameters have a high influence on the spatial and temporal distribution of runoff and are necessary to estimate the risk hazard (Gericke and Smithers, 2014). A detailed review of several methods to estimate hydrological response time was proposed by Gericke and Smithers (2014), with the aim to analyse the impact that time scale catchment parameters have on peak discharge.

The time of concentration $t_c$ is one of the most common hydrological characteristic time scales (Gericke and Smithers, 2014) and it is defined as the time that a drop that falls on the most remote part of the drainage basin needs to reach the basin outlet (Singh, 1997). The time of concentration is difficult to measure, because it assumes that initial losses are already satisfied and the rainfall event intensity is constant for a period at least as long as the time of concentration. Different theoretical definitions have been developed in order to estimate the time of concentration as function of basin length, slope and other characteristics (see for some examples (Singh, 1976; Morin et al., 2001; USDA, 2010; Gericke and Smithers, 2014).

Due to difficulties related to the estimation of time of concentration, Larson (1965) introduced the time of virtual equilibrium $t_{ve}$, defined as the time until response is 97% of runoff supply.

When a given rainfall rate persists on a region for enough time to reach the equilibrium, this time is called time to equilibrium $t_e$ (Ogden et al., 1995; Ogden and Dawdy, 2003; van de Giesen et al., 2005). Time of equilibrium for a turbulent flow on a rectangular runoff plane given rainfall intensity $i$, with given roughness $n$, length $L_p$ and slope $S$ can be written as (Ogden et al., 1995):

$$t_e = [\frac{nL_p}{S^{1/2}i^{2/3}}]^{3/5} \tag{2}$$

Another commonly used hydrological characteristic time scale is the lag time $t_{lag}$. It represents the delay between rainfall and runoff generation. $t_{lag}$ is defined as the distance between the hyetograph and hydrograph center of mass of (Berne et al., 2004), or between the time of rainfall peak and time of flow peak (Marchi et al., 2010; Yao et al., 2016). $t_{lag}$ can be considered characteristic of a basin, and is dependent on drainage area, imperviousness and slope (Morin et al., 2001; Berne et al., 2004; Yao et al., 2016). Berne et al. (2004), including the results of Schaake and Knapp (1967) and Morin et al. (2001), defined a relation between the dimension of the catchment area $S$ (in ha) and the lag time $t_{lag}$ (in mm): $t_{lag} = 3S^{0.3}$. Empirical relations between $t_{lag}$ and $t_c$ are presented in the literature (USDA, 2010; Gericke and Smithers, 2014),as, for example the approximation presented by Gericke and Smithers (2014), for which $t_{lag} = 0.6t_c$.

Another characteristic time scale is the 'response time scale' $T_s$, presented for the first time by Morin et al. (2001). It is defined as the time scale at which the pattern of the time averaged and basin averaged radar rainfall hyetograph is most similar





to the pattern of the measured hydrograph at the outlet of the basin. This definition was updated by Morin et al. (2002). Here, the response time was estimated using an objective and automatic algorithm, that analyses the smoothness of the hyetograph and hydrograph instead of the general behaviour. The smoothness is represented by the peak density $PD$, which is the ratio between the number of peaks $PN$ and total rising limbs duration $T_r$:

$$PD = \frac{PN}{T_r} \tag{3}$$

The peak density was estimated for the measured hyetograph and for the hydrographs, generated for rainfall measured with a C-band weather radar at a temporal resolution of 5 min, aggregate to a range of lower temporal resolutions. The response time scale was defined as the time scale corresponding to the aggregated temporal resolution, at which peak density of hydrograph and hyetograph matched closely. These works analysed the behaviour of four Israelian catchments. Results show that urban basins have a smaller $T_s$, about $10 - 15$ min, compared to semiarid rural regions, where $T_s$ was found between $125 - 130$ min. Morin et al. (2003) investigated also the effects of catchment characteristics, such as length and roughness of hillslope and channels, on the response time scale. For this study, a distributed model was used. They found that for small contributing areas $T_s$ is more affected by hillslope routing processes, while for larger areas, $T_s$ depends strongly on the channel flow length.

A similar definition of peak density was given by Shamir et al. (2005), who presented the rising limb density $RLD$ and declining limb density $DLD$, as ratio between peak number $PN$ and total duration of the rising $T_r$ and declining $T_d$ limbs of the hyetograph respectively:

$$RLD = \frac{PN}{T_r} \tag{4}$$

$$DLD = \frac{PN}{T_d} \tag{5}$$

A summary of time scale characteristics is presented in Tab. 1.

## 3 Rainfall measurement and variability in urban regions

Rainfall is an important driver for many hydrological processes and represents one of the main sources of uncertainty in studying hydrological response (Niemczynowicz, 1988a; Rico-Ramirez et al., 2015).

In this section instruments and technologies for rainfall measurement are described, pointing out their opportunities and and limitations for measuring spatial and temporal variability in urban environments. Subsequently, methods to characterise rainfall events according to their space and time variability are described.

### 3.1 Rainfall estimation

Rain gauges were the first instrument used to measure rainfall and are still commonly used, because they are relatively low in cost and easy to install (WMO, 2008).



Afterwards, weather radars were introduced to estimate the rainfall spatial distribution. These instruments allow to get measurements of rainfall spatially distributed over the area, instead of a point measurement as in the case of rain gauges. Rainfall data obtained from weather radars are used to study the hydrological response in natural watersheds and urban catchments (Einfalt et al., 2004; Berne et al., 2004; Sangati et al., 2009; Smith et al., 2013; Ochoa-Rodriguez et al., 2015b) often combined

with rainfall measurement from rain gauge networks (Winchell et al., 1998; Smith et al., 2005; Segond et al., 2007; Smith et al., 2012), as well as to improve short-term weather forecasting and nowcasting (Meselhe et al., 2008; Liguori and Rico-Ramirez, 2013; Dai et al., 2015).

More recently, commercial microwave links have been used to estimate the spatial and temporal rainfall variability (Fencl et al., 2015; Leijnse et al., 2007). Rainfall estimates are obtained from the attenuation of the signal caused by rain along

microwave link paths. This approach can be particularly useful in cities that are not well equipped with rain gauges or radars, but where the commercial cellular communication network is typically dense (Leijnse et al., 2007).

### 3.1.1 Rain gauges networks

Several types of rain gauges have been developed, such as weighing gauges, tipping bucket gauges and pluviographs. They are able to constantly register accumulation of rainfall volume over time, thus providing a measurement of temporal variability of

rainfall intensity. Rain gauge measurements are sensitive to wind exposure and the error caused by wind field above the rain gauge is $2 - 10\%$ for rainfall and up to $50\%$ for solid precipitations (WMO, 2008). Other errors can be due to tipping bucket losses during the rotation, to wetting losses on the internal walls of the collector, to evaporation (especially in hot climates) or water splashing into and out of the collector (WMO, 2008). The main disadvantage of rain gauges is that they measure rainfall in one specific location, and due to the high spatial variability of rainfall events, the measurement is often not representative

of a larger area. To solve the problem of spatial representation, interpolation techniques are used to obtain distributed rainfall fields (Shaghaghian and Abedini, 2013). Uncertainty induced by interpolation strongly depends on the density of the rain gauge network and on homogeneity of the rainfall field (Wang et al., 2015).

In urban areas, rainfall measurements with rain gauges present specific challenges associated with microclimatic effects introduced by the building envelope. WMO (2008) recommended minimum distances between rain gauges and obstacles of

one to two times the height of the nearest obstacle, a condition that is hard to fulfil in densely built areas. A second problem is introduced by hard surfaces, that may cause water splashing into the gauges. Rain gauges in cities are often mounted on roofs for reasons of space availability and safety from vandalism. This means they are affected by the wind envelope of the building, unless they are elevated to a sufficient height above the building.

Rain gauge measurement error can be $30\%$ or more depending on the type of instrument used for the measurement and local

conditions (van de Ven, 1990; WMO, 2008).

### 3.1.2 Weather radars

In the last decades, weather radars have been increasingly used to measure rainfall (Berne and Krajewski, 2013; F. and A., 2005; Niemczynowicz, 1988b; Otto and Russchenberg, 2011)). Radars transmit pulses of microwave signals and measure the





power of the signal reflected back by raindrops, snowflakes and hailstones (backscatter). Rainfall rate $R$ [L T$^{-1}$] is estimated using the reflectivity $Z$ [L$^6$ L$^{-3}$] measured from the radar through a power law:

$$R = aZ^b \qquad (6)$$

where $a$ and $b$ depend on type of precipitation, climate characteristics and spatial and temporal scales considered (van de Beek
et al., 2010; Smith et al., 2013). Weather radars of three different wavelengths $\lambda$, frequencies $\nu$ and sizes of the antenna $l$, are commonly used: S-band ($\lambda = 8 - 15$ cm, $\nu = 2 - 4$ GHz, $l = 6 - 10$ m), C-band ($\lambda = 4 - 8$ cm, $\nu = 4 - 8$ GHz, $l = 3 - 5$ m) and X-band ($\lambda = 2.5 - 4$ cm, $\nu = 8 - 12$ GHz, $l = 1 - 2$ m). X-band radars can be beneficial for urban areas: they are low cost and they can be mounted on existing buildings and measure rainfall closer to ground at higher resolution than national weather radar networks (Einfalt et al., 2004). Polarimetric weather radars transmit signals polarised in different directions (Otto
and Russchenberg, 2011), enabling it to distinguish between horizontal and vertical dimension, thus between rain drops and snowflakes as well as between smaller or larger more oblate rain drops.

A specific strength of polarimetric radars is the use of differential phase $K_{d}p$, which is not affected by attenuation, and is immune to radar calibration errors (Otto and Russchenberg, 2011; Ochoa-Rodriguez et al., 2015b).

### 3.1.3 Opportunities and limitations of weather radars and rain gauges

Berne and Krajewski (2013) presented a comprehensive analysis of the advantages, limitations and challenges in rainfall estimation using weather radars. One of the main problems is that the radar uses an indirect relation (Eq. (6)) to estimate rainfall and has to be adjusted based on rain gauges and disdrometers. Various techniques have been studied to calibrate radars (Wood et al., 2000), to combine radar rainfall measurements with rain gauge data for ground truthing (Cole and Moore, 2008; Smith et al., 2012; Wang et al., 2013; Gires et al., 2014; Wang et al., 2015) and to define the uncertainty related to radar-rainfall estimation (Mandapaka et al., 2009; Overeem et al., 2009a). These studies show that radar measurements underestimate the rainfall compared to rain gauge measurements (Smith et al., 2012; Overeem et al., 2009a; Overeem and Buishand, 2009b; van de Beek et al., 2010). For instance, Overeem et al. (2009a) compared rainfall measurements from two C-Band radars, located in the Netherlands, with two rain gauge networks, one with 326 manual rain gauges and another with 33 automatic rain gauges. They found that radar measurements underestimate rainfall by more than 30%, for 24h rainfall accumulations. They tested various techniques to reduce bias and standard deviation and found that hourly mean bias adjustment allowed to obtain zero bias error for some combinations of radar and rain gauge. Another downsides of radars is their installation at high locations to have a clear view without obstacles, while rainfall intensities can change before reaching the ground (Smith et al., 2012). Moreover, the radar equation can not account for variations in drop size distribution. Berne and Krajewski (2013) pointed out additional aspects that have to be taken into account like, e.g., management and storage of the high quantity of data that are measured, possibility to use the weather radars to estimate snowfall and the uncertainty related to it, and problems related to rainfall measurement in mountain areas.

High resolution rainfall data are particularly important for hydrological studies in urban areas, where dimensions of catchments are typically small and hydrological response is fast (Einfalt et al., 2004; Bruni et al., 2015). Rain gauge measurements in





urban areas tend to be prone to errors due to microclimatic effects introduced by the building envelope. In this context, the use of weather radar could represent a big improvement to obtain a more accurate rainfall information for studying hydrological response.

A promising application of radar is their combination with nowcasting models to obtain short-term rainfall forecasts. Liguori and Rico-Ramirez (2013) presented a review of different nowcasting models, that benefit from radar data. This work focused in particular on a hybrid model, able to merge the benefits of radar nowcasting and numerical weather prediction models. Radar data can provide an accurate short term forecast, but rainfall estimation is still affected by errors, and these limit the high-potential for the use in hydrological forecasting studies.

### 3.2 Influence of urban areas on rainfall

Several studies tried to estimate effects of increasing urbanization and changing land cover on rainfall (Huff and Changno, 1973; Shepherd et al., 2002; Shepherd, 2006). Increase in heat produced by human activities and changes in surface roughness, influencing rainfall and wind (Salvadore et al., 2015).

Shepherd et al. (2002) studied on 5 large metropolitan areas located in the United States, using data from the Tropical Rainfall Measuring Mission satellite's precipitation radar for the warm season (1998-2000). They found an average increase of about $28\%$ in monthly rainfall rates within $30 - 60km$ downwind of a metropolis, with a modest increase of $5.6\%$ over the metropolis. The maximum value was generally found at an average distance of $39km$ from the edge of the urban centre.

Smith et al. (2012) studied a 9 year radar rainfall data set, measured with resolution of $1x1$ km$^2$ and $15$ min respectively, for a $17000$ km$^2$ metropolitan area located in the Baltimora region(U.S.). They studied the effect of urbanisation on rainfall by comparing rainfall patterns upwind and downwind and found heavy precipitation above 25 mm daily accumulation to be four times more likely in the area downwind of the city compared to the area upwind.

Similar results were found by Daniels et al. (2015), at an at an extended urban region along the Dutch west coast. They used rainfall observations for the period 1951-2010 and found annual precipitation increase of about $7\%$ downwind of urban areas. This increase in rainfall was similar throughout the entire distribution of precipitation intensities, in extreme precipitation as well as the mean.

These studies show that areas seem to affect the local hydrological system, not only by increasing the imperviousness degree of the soil, but also by changing rainfall generation and intensity patterns.

### 3.3 Characterising rainfall events according to their spatial and temporal scale

Characterizations and classifications of intense rainfall events have been proposed by various authors. An example of characterisation of rainfall structure was given by Smith et al. (1994), who presented an empirical analysis of four extreme rainstorms in the Southern Plains (U.S.), using data from two networks of more than 200 rain gauges and from a weather radar. They defined *major rainfall event* as storms for which 25 mm of rain covered an area larger than $12500$ km$^2$. Thorndahl et al. (2014) presented a storm catalog of heavy rainfall, over a study area of $73500$ km$^2$ in southern Wisconsin, and key elements of storm evolution that control the scale. The catalog contains the 50 largest rainfall events recorded during a 16 year period



by WSR-88D radar with spatial and temporal resolution of 1x1 km$^2$ and 15 min respectively. Over the 50 events, there is 0.60 probability that rainfall exceeds 25 mm of daily accumulation in a 1 km$^2$ pixel and 0.14 probability of exceeding 100 mm. Results showed that there is a clear relation between the characteristic length and time scale of the events. The length scale increased with time scale: a length scale of $35 \pm 20$ km was found for a time step of 15 minutes, up to $160 \pm 25$ km for a 12 hour aggregation time.

Studying rainfall variability at the urban scale, Emmanuel et al. (2012) classified 24 rain periods, recorded by the weather radar located in Treillieres (France), with a spatial and temporal resolution of $250 X 250$ m$^2$ and 5 min respectively. They classified the events into four groups, based on variogram analysis: light rain period, shower periods, storms organized into rain bands and unorganized storms. The first group, characterized by light rainfall events, presented very high decorrelation distance and time (17 km and 15 min) compared to the second group, with a decorrelation distance and time of 5 km and a decorrelation time of 5 min. The last two groups presented a double structure, where small and intense clusters, with low decorrelation distance and time (less than 5 km and 5 min) are located, in a random or organized way, inside areas with a lower variability (decorrelation of 15 km and 15 min).

Jensen and Pedersen (2005) presented a study about variability in accumulated rainfall within a single radar pixel of 500x500 m$^2$, comparing it with 9 rain gauges located in the same area. The results showed a variation of up to $100\%$ at a maximum distance of about 150 m, due to the rainfall spatial variability. This study suggested that a huge quantity of rain gauges is needed to have a powerful rain gauge network capable of representing small scale variability.

Gires et al. (2014) focused on the gap between rain gauges and radar spatial scale, considering that a rain gauge usually collects rainfall over 20 cm of surface and the spatial resolution of most used radars is of 1x1 km$^2$. They evaluate the impact of small scale rainfall variability using a Universal Multifractal downscaling method. The downscaling process was validated with a dense rain gauge and disdrometer network, with 16 instruments located in 1x1 km$^2$. They showed two effects of small scale rainfall variability that are often not taken into account: high rainfall variability occurred below 1 km$^2$ spatial scale and the random position of the point measurement within a pixel influenced measured rainfall events.

In Tab. 2 four types of rainfall events are presented with their characterization and typical spatial and temporal decorrelation lengths, based on van de Beek et al. (2010); Emmanuel et al. (2012); Smith et al. (1994). Considering that the minimal rainfall measurement resolution required for urban hydrological modelling is 0.4 the decorrelation length (Julien and Moglen, 1990; Berne et al., 2004; Ochoa-Rodriguez et al., 2015b), common operational radars are not able to satisfy this requirement.

## 4   Hydrological processes

In this section, general characteristics and parametrisations of hydrological processes are presented, highlighting their spatial and temporal variability and characteristics specific to urban environments.





### 4.1 Precipitation losses

#### 4.1.1 Infiltration, interception and storage

The term infiltration is usually used to describe the physical processes by which rain enters the soil (Horton, 1933). Richards equation (Richards, 1931) describes infiltration using a partial differential equation with nonlinear coefficients. The Richards

equation combines Darcy's law, continuity equation and the definition of the total head:

$$\frac{\partial \theta}{\partial t} = -\frac{\partial q}{\partial z} = \frac{\partial}{\partial z}(D\frac{\partial \theta}{\partial z} + K) \qquad (7)$$

where $\theta$ [L$^3$ L$^{-3}$] is water content, $t$ [T] is time, $q$ [L T$^{-1}$] is infiltration flow, $z$ [L] is depth, $D$ [L T$^{-1}$] is diffusivity and $K$ [L T$^{-1}$] is hydraulic conductivity. Several models and equations have been proposed to solve the Richards equation in order to estimate infiltration rate. The two terms Philip's equation (Philip, 1957), for example, assumes that hydraulic conductivity $K$

and diffusivity D are functions of water content $\theta$, and they do not depend on $z$. With these assumptions, Eq. (7) becomes:

$$I(t) = St^{-\frac{1}{2}} + K \qquad (8)$$

where $t$ is time, $S$ [L T$^{-1}$] is sorptivity and $K$ is hydraulic conductivity. The sorptivity $S$ is defined as capacity of soil to absorb water by capillarity.

Another possibility to estimate the infiltration capacity is given by the empirical equation presented by Horton (1933). In

Horton's equation hydraulic conductivity $K$ and diffusivity $D$ are constants, and do not depend on water content $\theta$ or on depth $z$. The infiltration capacity $f_t$ decreases exponentially with $t$, following the relation:

$$f_t = f_e + (f_b - f_e)e^{-k_a t} \qquad (9)$$

where $f_b$ [L T$^{-1}$] and $f_e$ [L T$^{-1}$] are the maximum and minimum infiltration capacity respectively, and $k_a$ [T$^{-1}$] is the time factor for decreasing infiltration capacity (Horton, 1939).

If water cannot infiltrate, as is the case in impervious areas, it can be stored in local depressions, where it does not contribute to runoff flow. This is the case of local depressions on streets or flat roofs, where water accumulates until the storage capacity is reached. Before reaching the ground, rainfall can be intercepted by vegetation cover or buildings. Interception can constitute up to 20% of rainfall at the start of a rainfall event (Mansell, 2003), and decreases quickly to zero, once surfaces are wetted.

Spatial scale of precipitation losses is strongly influenced by land cover variation. In urban areas, land cover variability

typically occurs at a spatial scale of 100 m to 1000 m. Time scale is associated with local storage accumulation volume, sorptivity and hydraulic conductivity, which in turn depend on soil type and soil compaction.

In impervious areas, water can not infiltrate, but can be stored in local depressions, where it does not contribute to runoff flow. This is the case of local depression on street or flat roofs, where water accumulates till the storing capacity is reached.

#### 4.1.2 Infiltration and subsurface processes in urban areas

Groundwater recharge mechanisms change due to human activities and urbanization, both in terms of volume and quality of the water. The increase of imperviousness of land cover leads to a decrease in infiltration of rainfall into soil, reducing





direct recharge of groundwater. The presence of leakage from drinking water and sewer networks can increase infiltration to groundwater and amount of contaminants that is spread from the sewer system into the soil (Salvadore et al., 2015).

Although it is well known that not all the rainfall turns into runoff (Boogaard et al., 2013; Lucke et al., 2014), it is common to consider the losses from impervious areas so small that they can be assumed negligible compared to the total runoff volume

(Ragab et al., 2003; Ramier et al., 2011). Some studies tried to emphasise the importance of accounting for the infiltration in the study of the urban water balance, considering that infiltration through the road surface can constitute between 6 and 9% of annual rainfall (Ragab et al., 2003). Due to high spatial variability of infiltration, representative measurements are difficult to obtain and require a large amount of point-scale measurements (Boogaard et al., 2013; Lucke et al., 2014).

Several types of pervious pavements are used in urban areas. Monolithic structures consist of a combination of impermeable

blocks of concrete and open joints or apertures, that allow water to infiltrate. Modular structures, gaps between two blocks are not filled with sand, as with conventional pavements, but with $2-5$ mm of bedding aggregate, that facilitate infiltration (Boogaard et al., 2013). Following European standards, minimum infiltration capacity for permeable pavements is 270 $l\,s^{-1}\,ha^{-1}$, equal to 97.2 mm $h^{-1}$ (OCW, 2008). The effects of age on the efficiency of permeable pavements was analysed by Boogaard et al. (2013), who studied 55 permeable pavements with ages between 1 and 12 years, located in the Netherlands

and in Australia. They showed that the performance of clogged permeable pavement system was higher than the required rate in more than 90% of the cases.

Pervious areas in cities can effectively act as semi−impervious areas, because within the soil column there is a shallow layer that presents a low hydraulic conductivity at saturation, caused by soil compaction during the building process. (Smith et al., 2015) studied the influence of this phenomenon on peak runoff flow by applying 21 storm events on a physically based,

minimally calibrated model of the Dead Run urban area (U.S.) with and without the compacted soil layer. Results showed that the compacted soil layer reduced infiltration by $70-90\%$ and increased peak discharge by $6.8\%$

## 4.2 Surface runoff

When rainfall intensity exceeds infiltration capacity of the soil, water starts to accumulate on the surface and flows following the slope of the ground. This process is generally called Hortonian runoff (Horton, 1933) or infiltration capacity excess flow.

It is usually contrasted with saturation excess flow, or Dunne flow (Dunne, 1978), that occurs when the soil is saturated and rainfall can no longer be stored (van de Giesen et al., 2011). Different numerical models were built to describe overland flow generated during rainfall events (Ajayi et al., 2008; van de Giesen et al., 2011). These models are mainly based on the de Saint Venant equations (Eq. (10) and Eq. (11)), which are the monodimensional expression of the Navier-Stokes equations for shallow flow, where vertical flow is much smaller than horizontal flow and can be neglected.

In urban areas, runoff is generated when the surface is impervious and water can not infiltrate, or when infiltration capacity is exceeded by rainfall intensity. Water flows over the surface and can reach natural drainage channels or be intercepted by the drainage network through gullies and manholes. If the drainage network capacity is exceeded, the system become pressurized, and water starts to flow out from gullies, increasing runoff on the street (Ochoa-Rodriguez et al., 2015a).





It is important to pay attention to some elements that characterize the runoff in urban environments: sharp corners or obstacles can, for example, deviate the flow and introduce additional hydraulic losses. Runoff flows are often characterised by very small water depths that are often alternated with dry surfaces, especially when rainfall intensities vary strongly in space and time.

### 4.2.1 Approximation of the de Saint Venant equations

The de Saint Venant equations are expressed by a conservation mass equation and a momentum conservation or dynamic equation. They can be written as:

$$\frac{\partial h}{\partial t} + u\frac{\partial h}{\partial x} + h\frac{\partial u}{\partial x} - q = 0 \tag{10}$$

$$\frac{\partial u}{\partial t} + \frac{\partial u}{\partial x} + g\cos\beta\frac{\partial h}{\partial x} - g(S_0 - S_f) - \frac{q_u}{h} = 0 \tag{11}$$

where $x$ [L] is the direction of the one dimensional flow, $t$ [T] is the time, $h$ [L] is the depth, $u$ [L T$^{-1}$] is the average velocity at $(x,t)$ and $q$ [L T$^{-1}$] is the lateral inflow per unit area per unit time, $g$ [L T$^{-2}$] is the gravity acceleration, $\beta$ the constant angle of the slope, and $S_f$ [-] the friction slope (Daluz Veira, 1983).

Daluz Veira (1983) defined the conditions to approximate the de Saint Venant equations for shallow surface water, considering different values of the Froude number Fr, and the kinematic wave number k. The Froude number $F_r$ [-] is a dimensionless

quantity expressed by $F_r = \frac{v}{\sqrt{gy}}$ where $v$ [L T$^{-1}$] is the velocity of the flow, $g$ [L T$^{-2}$] is gravity acceleration and $y$ [L] is a characteristic length. The kinematic wave number $k$ is defined by $k = L\,S_0/F_0^2\,y_0$, where $L$ [L] is the length of overland-flow plane or channel segment, $S_0$[-] is the bed slope, $F_0$ [-] is the Froude number for normal flow and $y_0$ [L] is normal depth. For values of k higher than 50, it is possible to simplify the momentum equation, and obtain the kinematic wave approximation, that does not account for the advective acceleration. For high values of $k$ and $Fr < 0.1$, the diffusion wave approximation, that

neglects inertial acceleration, is obtained, while the gravity wave approximation is defined for small values of k. For example, for smooth urban slopes, the values of k is usually between 5 and 20: the kinematic or diffusion wave approximation may be used, depending on the value of Fr.

### 4.2.2 Impact of land cover on overland flow in urban areas

Ragab et al. (2003) presented an experimental study of water fluxes in a residential area, in which they estimated infiltration

and evaporation in urban areas, showing that the assumption that all rainfall becomes runoff is not correct and that it leads to an overestimation of runoff. Ramier et al. (2011) studied the hydrological behaviour of urban streets over a 38-month period to estimate runoff losses and to better define rainfall runoff transformations. They estimated losses due to evaporation and infiltration inside the road structure between 30 and $40\%$ of the total rainfall.

The impact of increase of imperviousness on hydrological response was studied by Cheng (2002), who analysed the effects

of urban development in Wu-Tu (Taiwan's catchment) considering 28 rainfall events (1966-1997). Results showed that response





peak increased by 27% and the time to peak decreased from 9.8 to 5.9 hours, due to an increase of imperviousness from 4.78% to 11.03%.

In a similar study, Smith et al. (2002) analysed the effects of imperviousness on flood peak in the Charlotte metropolitan region (U.S.), analysing a 74 year discharge record. Results showed that different land covers were associated with large
differences in timing and magnitude of flood peak, while there were not significant differences in the total runoff volume. Hortonian runoff was the dominant runoff mechanism. Antecedent soil moisture is playing an important role in this watershed, even in the most urbanized catchment, where the effects are generally less evident.

The influence of antecedent soil moisture is, however, not always so evident. Smith et al. (2013) showed that in nine watersheds, located in the Baltimore metropolitan area, the antecedent soil moisture, defined as 5 day antecedent rainfall, seemed
not to affect the hydrological response. Introduction of stormwater management infrastructure played an important role in reducing flood peaks and increasing runoff ratios. Results showed that rainfall variability may have important effects on spatial and temporal variation in flood hazard in this area.

Analysing the effects of a moderate extreme and an extreme rainstorm on the same area presented by Smith et al. (2013), Ogden et al. (2011) highlighted the importance of changes in imperviousness on flood peaks. For extreme rainfall event,
imperviousness has a small impact on runoff volume and runoff generation efficiency.

### 4.3 Evaporation

Evaporation plays an important role in the hydrological cycle: in forested catchment around $60-95\%$ of total annual rainfall evaporates or is absorbed by the vegetation (Fletcher et al., 2013). In an urban catchment, effects of the evaporation are drastically reduced (Oke, 2006; Fletcher et al., 2013; Salvadore et al., 2015). Evaporation is often neglected in analysis of
fast and intense rainfall events: the order of magnitude of evaporation is very small compared to the total amount of rainfall. Recent studies have shown that evaporation is not always negligible in urban areas and can constitute up to $40\%$ of the annual total losses (Grimmond and Oke, 1991; Salvadore et al., 2015). In their experimental study, Ragab et al. (2003) showed that evaporation represents, $21-24\%$ of annual rainfall, with more evaporation taking place during summer than winter. It is particularly important to have measurements with high resolution because a coarse spatial description can hide heterogeneous
land covers and consequently, heterogeneous evaporation losses (Salvadore et al., 2015). Different techniques and approaches have been developed to measure the impact of evaporation, from the standard lysimeter to the use of remote sensing (Nouri et al., 2013).

Evaporation measurements in urban areas are one of the weak points of the water balance (van de Ven, 1990) and they present many problems and challenges (Oke, 2006). It is quite hard in fact to find a site, representative of the area, that is far
enough from obstacles, not placed on concrete or asphalt and not unduly shaded. Errors in estimation of annual evaporation may still be higher than 20% (van de Ven, 1990).



## 4.4 Flow in sewer systems

In urban areas, part of the surface runoff enters in the sewer system through gully inlets, depending on the capacity of these elements, on their maintenance (Leitão et al., 2016) and the sewer system itself.

Stormwater flow in sewer systems is highly non-uniform and unsteady, it can be considered as one dimensional, assuming that depth and velocity vary only in the longitudinal direction of the channel. Flow in sewer pipes is usually free-surface, but during intense rainfall events the system can become full and temporarily behave as a pressurised system, a phenomenon called surcharge. In particular conditions, as for example in flat catchments, inversion of the flow direction in pipes can occur during filling and emptying of the system. The most common form to model flow in sewer pipes is based on a one-dimensional form of the de Sain-Venant(Eq. (10) and Eq. (11)).

Sewer system density influences runoff generation (Ogden et al., 2011; Yang et al., 2016): a dense pipe network can, in fact, reduce the runoff generation, increasing the storage capacity of the system (Yang et al., 2016). Ogden et al. (2011)presented a study about the importance of drainage density on flood runoff in urban catchments. Defining the drainage density as channel length per total catchment area, they studied the hydrological response of the same basin modelled with drainage density that varied from 0.4 km km$^{-2}$ and 3.9 km km$^{-2}$. Results showed a significant increase in peak discharge and runoff volume for drainage density between 0.4 km km$^{-2}$ and 0.9 km km$^{-2}$, while for values higher than 0.9 km km$^{-2}$, effects were negligible. When the storage and transport capacity of a system is not sufficient to prevent flooding, detention basins are effective tools to reduce peak flows, and they can reduce the superficial runoff up to 11% (Smith et al., 2015).

Similarly, green roofs can significantly decrease and slow peak discharge and reduce runoff volume. Versini et al. (2014) presented a study on the impact of green roofs at urban scale using a distributed rainfall model. They showed that green roofs can reduce runoff generation in terms of peak discharge, up to 80% depending on the rainfall event and initial conditions.

## 5 Urban hydrological models

Urban hydrological models were developed since the 1970s to better understand the behaviour of the components of the water cycle in urban areas (Zoppou, 2000). Since then, many models, with different characteristics, principles and complexity have been built. These models are used for several purposes, such as to study and predict the effects of urbanization increase on the hydrological cycle, to support flood risk management, to ensure clean and fresh drinking water for the population, and to support improvement of waste water networks and treatments.

Hydrological models have shown to be useful to compensate partially for the lack of measurements (Salvadore et al., 2015), but all models present errors and uncertainties of different nature and magnitude (Rafieeinasab et al., 2015). In this chapter, different classifications and characterizations of hydrological models are presented.





## 5.1 Urban hydrological model characterization

A first distinction that can be made is between deterministic and stochastic models. They differ in the use of random variables to solve mathematical relationships that describe system behaviour. Stochastic models use random values from a chosen probability distribution for one or more model parameters or input variables. Deterministic models do not consider any randomly chosen values and will always produce the same identical results for a given configuration of variables. For given inputs, stochastic models account for the uncertainty in input variables and model parameters, by incorporating selected probability distributions.

Models can also be classified according to the representation of spatial variability of the catchment. A lumped model does not consider spatial variability of the input, and uses spatial averaging to represent catchment behaviour. In contrast, distributed models describe spatial variability, usually using a node-link structure to describe subcatchment components (Zoppou, 2000; Fletcher et al., 2013). A general criterion to choose between lumped and distributed models was presented by Berne et al. (2004). The representative surface associated to a single rain gauge $S_r$ [L$^2$], defined in relation to the rainfall spatial resolution $r$ as $S_r = \pi[r/2]^2$, was compared with the surface area of a catchment $S$ [L$^2$]. If $S_r > S$ or $S_r \sim S$ a lumped modelling approach was suggested, while for $S_r < S$, they recommended using a distributed model, as well as collecting measurements at the subcatchment scale. Different sub-categories are presented to characterize model spatial variability. Distributed models can be divided into fully distributed and semi-distributed models. Fully distributed models present a detailed discretization of the surface using a grid or a mesh of regular or irregular elements, and apply the rainfall input to each grid element, generating grid-point runoff. Semi-distributed models are based on subcatchment units, through which rainfall is applied. Each subcatchment is modelled in a lumped way, with uniform characteristics and a unique discharge point (Pina et al., 2014). Salvadore et al. (2015) proposed a model classification based on spatial variability with 5 categories: lumped, semi-distributed, Hydrological Response Unit based (semi-distributed with a specific way to define the subcatchment area), grid based spatially distributed and Urban Hydrological Element based (mainly focused on the urban fluxes).

Another distinction is between conceptual and physically based (or process based) models, depending on whether the model is based on physical laws or not. Recently, Fatichi et al. (2016) presented an overview of the advantages and limitations of physically based models in hydrology. They defined a physically based hydrological model as "a set of process descriptions that are defined depending on the objectives". The downsides of using a physically based model are related to over−complexity and over−parametrization: conceptual models are much easier to manage and they are usually less affected by numerical instability. Physically based models usually require high computational power and time and a large number of parameters, but there are situations in which it is important to keep the complexity to better understand the system mechanisms. They are also necessary to deal with the system variability and they allow to include a stochastic component to represent the uncertainty in parameter and input values.





## 5.2 Spatial and temporal variability in urban hydrological models

Spatial variability of land cover and soil characteristics is an important element in hydrological models, especially at the small urban scale. Choosing between a lumped, semi-distributed or fully distributed hydrological model leads to different representation of catchment characteristics and, consequently, to a different output.

A comparison between semi-distributed and fully-distributed urban stormwater models was made by Pina et al. (2016). Two small urban catchments, Cranbrook (London, UK) and the centre of Coimbra (Portugal), were modelled with a semi- and a fully-distributed model. Flow and depth in the sewer system of the different models were compared with observations and, in general, semi-distributed models predicted sewer flow patterns and peak flows more accurately, while fully distributed models had a tendency to underestimate flows. This was mainly due to the presence small−scale surface depressions, building
singularities or lack of representation of private connections. Although fully-distributed models are more realistic and able to better represent spatial variability of the land cover, they need a higher resolution and accuracy to define module connections. Calibration of detailed, distributed models remains a complex issue that is not yet well resolved. The authors suggested to use a semi-distributed model approach in cases of low data resolution and accuracy.

    To study the hydrological response Aronica and Canarozzo (2000) presented the UDTM, a model that represents sub-
catchments of a semi−distributed model with two conceptual linear elements: a reservior and a channel. In a more recent study (Aronica et al., 2005), this model was compared to the EPA SWMM model, that allows the user to choose different conceptual models to simulate runoff and sewer flow. Results showed that model structure and sensitivity to parameters influence the sensitivity to the rainfall input resolution.

    Depending on their characteristics, models can be very sensitive to spatial and temporal rainfall variability or not be able
to correctly reproduce their effects. In the literature, different comparisons between models and their sensitivity to spatial and temporal rainfall resolution have been presented. An example is given by Meselhe et al. (2009), who investigated the impact of temporal and spatial sampling of rainfall on runoff predictions using a physically based [System Hydrologique European (MIKE SHE)] and conceptual [hydrologic modelling system (HMS)] hydrologic models. Their study showed that the physically based model was more sensitive to both spatial and temporal rainfall samplings than the conceptual model.

## 6   Interaction of spatial and temporal rainfall variability with hydrological response in urban basins

Storm structure and motion play an important role in the variabilty of the hydrological response (Smith et al., 1994; Bacchi and Kottegoda, 1995; Ogden et al., 1995; Singh, 1997; Emmanuel et al., 2012; Nikolopoulos et al., 2014; Emmanuel et al., 2015), especially for small catchments (Faures et al., 1995; Fabry et al., 1994). The characterization and the influence of spatial and temporal rainfall variability on runoff response is still not well understood (Emmanuel et al., 2015).

Recent studies address the impact of rainfall variability, focusing on urban catchments (Berne et al., 2004; Ochoa-Rodriguez et al., 2015b; Rafieeinasab et al., 2015; Yang et al., 2016). The main results and conclusions are presented in the following sections. It is discussed how basin characteristics impact the sensitivity of hydrological response to rainfall variability and how the interaction between spatial and temporal rainfall variability influences hydrological response.





## 6.1 Interaction between rainfall resolution and urban hydrological processes

Many studies highlight the importance of high resolution rainfall data (Notaro et al., 2013; Emmanuel et al., 2012; Bruni et al., 2015) and how their use could improve runoff estimation, especially in an urban scenario, where drainage areas are small and spatial variability is high (Schilling, 1991; Schellart et al., 2011; Smith et al., 2013).

A theoretical study, conducted by Schilling (1991), emphasised the necessity to use rainfall data with a higher resolution for urban catchments compared to rural areas, and suggested to choose a minimum temporal resolution of $1 - 5$ min and a spatial resolution of $1$ km. The effects of temporal and spatial rainfall variability below $5$ min and $1$ km scale were subsequently studied by Gires et al. (2012). They investigated the urban catchment of Cranbrook (London, UK), with the aim of quantifying uncertainty in urban runoff estimation associated with unmeasured small scale rainfall variability. Rainfall data were obtained from the national C-band radar with a resolution of $1\,\mathrm{km}^2$ and $5$ min and were downscaled with a multifractal process, to obtain a resolution $9-8$ times higher in space and $4-1$ in time. Uncertainty in simulated peak flow associated with small-scale rainfall variability was found to be significant, reaching $25\%$ and $40\%$ respectively for frontal and convective events.

Required rainfall resolution for urban hydrological modelling strongly depends on the characteristics of the catchment. Several researchers have studied the sensitivity of urban hydrological response to different rainfall resolutions, highlighting correlations between rainfall resolution and catchment dimensions, such as drained area (Berne et al., 2004; Ochoa-Rodriguez et al., 2015b) or catchment scale length (Ogden and Julien, 1994; Chirico et al., 2001; Bruni et al., 2015).

### 6.1.1 Influence of spatial and temporal rainfall variability in relation with drained area

Drainage area dimensions influence hydrological response and their sensitivities to spatial and temporal rainfall resolution have recently been investigated.

Berne et al. (2004) studied the hydrological response of six urban catchments located in the south-east of the French Mediterranean coast. Rainfall data and runoff measurements were collected using two X-band weather radars, one vertically pointing radar and one radar performing vertical plane cuts of the atmosphere, with a spatial resolution of 7.5 m and 250 m and a temporal resolution of 4s and 1min respectively. The minimum temporal resolution required $\Delta t$ [T] was defined as $\Delta t = \frac{t_c}{4}$, where $t_c$ [T] is the characteristic time of a system and the value 4 depends on catchment properties (Schilling, 1991). By considering lag time $t_{lag}$ equal to the characteristic time $t_c$, it was possible to write the minimum required temporal resolution as a function of surface area $S$, based on the relationship $t_{lag} = S^{0.3}$: $\Delta t = 0.75\,S^{0.3}$. Spatial resolution was studied considering rainfall data collected from the X-band weather radar performing vertical plane cuts of the atmosphere, combined with measurements of rain gauges. Two spatial climatological variograms were built with a time resolution of 1 min (from radar) and 6 min (from a network of 25 rain gauges). Based on variogram analysis, it was possible to define the relation between range $r$ and time resolution $\Delta_t$ as: ($r = 4.5\sqrt{\Delta t}$). The minimum required spatial resolution $\Delta s$ [L$^2$] was defined by the author as $\Delta s = \frac{r}{3}$, and it can also be expressed as a function of $\Delta t$:

$$\Delta s = 1.5\Delta t. \tag{12}$$



In this way, both spatial and temporal resolution requirements were defined as a function of surface dimensions of a catchment. Required resolutions for urban catchments of 100 ha are 3 min and 2 km, but common operational rain gauge networks are usually less dense, while radars seldom provide data at this temporal resolution. Results presented are valid for catchments with characteristics similar to the catchments studied, such as surface area (from 10 ha to 10000 ha), slope (1% to 10%),

imperviousness degree (10% to 60%), and exposed to climatic conditions similar to those of Mediterranean area.

Ochoa-Rodriguez et al. (2015b) analysed the impact of spatial and temporal rainfall resolution on hydrological response in seven urban catchments, located in areas with different geomorphological characteristics. Using rainfall data measured by a dual polarimetric X-band weather radar with spatial resolution of $100 \times 100$ m$^2$ and temporal resolution of 1min, they investigated the effects of combinations of different resolutions, with the aim to identify critical rainfall resolutions. A strong relation

between drainage area and critical rainfall resolution and between spatial and temporal resolutions was found. Sensitivity to different rainfall resolutions decreased when the size of the subcatchment considered increased, especially for catchment size above 1 km$^2$. This study highlighted the importance of high resolution rainfall data as input. Spatial resolution of $3 \times 3$ km$^2$ is not adequate for urban catchments and temporal resolution should be lower than 5 min. Most operational radars present a temporal resolution of 5 min, not sufficient to correctly represent the effects of temporal rainfall variability.

The sensitivity to rainfall variability on 5 urban catchments of different sizes, located in the City of Arlington and Grand Prairie (U.S.), was studied with a distributed hydrological model (HLRDHM, Hydrology Laboratory Research Distributed Hydrological Model) by Rafieeinasab et al. (2015). Rainfall data were provided by the Collaborative Adaptive Sensing Atmosphere (CASA) X-band radar with spatial resolution of $250 \times 250$ m$^2$ and temporal resolution of 1 minute and upscaled in various steps to $2 \times 2$ km$^2$ and 1 hour. Results showed peak intensity and time to peak error to be sensitive to spatial rainfall

variability. The model was able to represent observed variability for all catchments except the smallest (3.4 km$^2$) at a temporal resolution of 15 minutes or lower, combined with spatial variability of $250 \times 250$ m$^2$ and capture variability in streamflow.

Rainfall required resolutions is higher for small basins, as in the case of urban catchments. The influence of slope, imperviousness degree or soil type were not separately investigated, but the relationships between catchment area and rainfall resolution are expected to depend on these characteristics as well.

**6.1.2   Influence of spatial and temporal rainfall variability in relation with length scale**

Sensitivity of hydrological response to different spatial and temporal rainfall resolutions have been investigated with dimensionless parameters to represent the length scales of storm events, catchments and of sewer networks.

Ogden and Julien (1994) identified dimensionless parameters to analyse correlations between catchment and storm characteristics and to study sensitivity of runoff models to radar rainfall resolution. Rainfall data of a convective storm event,

measured by a polarimetric radar with a spatial resolution of $1 \times 1$ km$^2$, were applied on two basins. The storm smearing was defined as the ratio between rainfall data grid size and rainfall decorrelation length. Storm smearing occurs when rainfall data length is equal or longer than the rainfall decorrelation length. The watershed smearing was described as the ratio between rainfall data grid size and basin length scale. When infiltration is negligible, watershed smearing is an important source of hydrological modelling errors, if the watershed ratio (rainfall measurement length/basin length) is higher than 0.4.





A similar approach, with dimensionless parameters, was recently applied by Bruni et al. (2015) to urban catchments. Rainfall data from a X-band dual polarimetric weather radar were applied to an hydrodynamic model, to investigate sensitivity of urban model outputs to different rainfall resolutions. Runoff sampling number was defined as ratio between rainfall length and runoff area length. Results confirm what was found by Ogden and Julien (1994). A third dimensionless parameter, called runoff

sampling number, was identified. Small-scale rainfall variability at the $100 \times 100$ m$^2$ affects hydrological response and the effect of spatial resolution coarsening on rainfall values strongly depends on the movement of storm cells relative to the catchment.

Using dimensionless parameters is a productive approach to study sensitivity of hydrological response to spatial and temporal rainfall variability. Effects of other catchment characteristics, such as slope or imperviousness, were so far neglected, but they need a deeper investigation.

## 10   6.2   Spatial vs temporal resolution

As it was already discussed in previous sections, there is a dependency between spatial and temporal rainfall required resolution and they affect in a different way the hydrological response (Marsan et al., 1996; Singh, 1997; Berne et al., 2004; Gires et al., 2011; Ochoa-Rodriguez et al., 2015b).

A first interaction between spatial and temporal rainfall scale was defined assuming that atmospheric properties are valid

also for rainfall. Following this assumption, Kolgomorov's theory (Kolgomorov, 1962) was combined with the scale invariance property of Navier-Stokes equation, in order to define the anisotropy coefficient $H_t$ (Marsan et al., 1996; Deidda, 2000; Gires et al., 2011). Temporal and spatial scale changing law ($t_\lambda$ and $s_{lambda}$) are defined using scaling factors ($\lambda_t$ and $\lambda_s$), related by the anisotropy coefficient $H_t$:

$$t_\lambda \mapsto t_\lambda / \lambda^{1-H_t} \tag{13}$$

$$s_\lambda \mapsto s_\lambda / \lambda \tag{14}$$

$$\lambda_t = \lambda_s^{1-H_t} \tag{15}$$

$H_t$ is a priori unknown for rainfall, but it can be assumed equal to $1/3$, a value that characterise atmospheric turbulence (Marsan et al., 1996; Gires et al., 2011, 2012). Lovejoy and Schertzer (1991) estimated $H_t = 0.5 \pm 0.3$ for raindrops.

Studying the hydrological response of the south-east French Mediterranean coast, Berne et al. (2004) proposed another

relationship between spatial $\Delta_s$ and temporal $\Delta_t$ resolution used to measure rainfall, as : $\Delta s = 1.5\sqrt{\Delta t}$ (see section 6.1.1 for the formula derivation).

Ochoa-Rodriguez et al. (2015b) derived the theoretically required spatial rainfall resolution for urban hydrological modelling starting from a climatological variogram, that characterised average spatial structure of rainfall fields over the peak storm period, fitted with an exponential variogram model. They defined characteristic length scale $r_c$ [L] of a storm event as

$r_c = (\frac{\sqrt{2\pi}}{3})r$, where $r$ [L] is the variogram range. The minimum required spatial resolution for adequate modelling of urban





hydrological response was defined as half characteristic length scale of the storm:

$$\Delta s = \frac{r_c}{2} \cong 0.418r. \tag{16}$$

The theoretically required temporal resolution $\Delta t$, was defined based on the time needed for a storm to move over distance equal to the characteristic length scale of the storm event $r_c$. It can be written as:

$$5 \quad \Delta t = \frac{r_c}{v}, \tag{17}$$

where $v$ [L t$^{-1}$] is the magnitude of the mean storm velocity, obtained from average of the velocity vectors (magnitude and direction) estimated at each time step. Authors investigated the impact of 16 combinations of 4 different spatial resolutions (100x100 m, 500$x$500 m, 1000x1000 m, and 3000x3000 m) combined with 4 different temporal resolutions (1, 3, 5 and 10 min). Resolution combinations were chosen considering different aspects, such as the operational resolution of radar and rain gauges networks, characteristics temporal and spatial scale already discussed in the literature (Berne et al., 2004), and according to Kolgomorov's scaling theory (Kolgomorov, 1962). Results showed that hydrodynamic models are more sensitive to the coarsening of temporal resolution of rainfall inputs than to the coarsening of spatial resolution, especially for fast moving storms. Critical rainfall resolutions were identified, considering the drainage area (Tab. 3). For small catchments, with area smaller than 1 ha, was found to be equal to 100x100 m and 1 min, while for areas between 1 ha and 100 ha, a spatial resolution of 500x500 m can be sufficient to estimate the hydrological response. The critical spatial resolution found is lower than 5 min, for catchment size from about 250 to 900 ha. Results were confirmed by Yang et al. (2016), that presented an analysis of flash flooding in two small urban subcatchments of Harry's Brook (Princeton, New Jersey, US), focusing on the influence of rainfall variability of storm events on hydrological response.

Moreover, spatial variability seemed to influence timing of runoff hydrograph, while temporal variability mainly influences peak value Singh (1997).

These studies highlighted the relatively more important role of temporal variability compared to spatial variability, for extreme rainfall events. The impact of the spatial variability, seemed to decrease with increase of total rainfall accumulation.

## 7 Discussion

In this article, the state of the art of spatial and temporal variability impact of rainfall and catchment characteristics on hydrological response in urban areas has been presented.

A first aspect that has been analysed is the high variability in space and time of hydrological processes and phenomena, highlighting how difficult it is to define spatial and temporal scale parameters, that are able to characterize catchments in an effective way. Several definitions to classify time scale characteristics are available in the literature, such as time of concentration, lag time, time of equilibrium and response time scale. However, measurement or estimation of those parameters is often difficult, which implies a high level of uncertainty. For this reason, thus far, no common agreement has emerged on a unique set of parameters able to characterize the variability of a catchment.



A similar problem can be observed for rainfall analysis, where there is not an unique way to classify rainfall events and to consider the spatial and temporal variability. In the last decades, new technologies have been developed in order to reduce the uncertainty connected to measurements of rainfall and to capture its variability. Weather radars are a good example of recently developed instruments, able to estimate rainfall spatial variability and to reduce uncertainty, especially when combined with

rain gauge networks. High resolution rainfall data are necessary to estimate the hydrological response, especially in urban areas, where rainfall effects are combined with a high variability of catchment characteristics and hydrological processes, such as infiltration, evaporation and surface runoff. An important role in urban areas is played by drainage infrastructures that highly affect the hydrological response, while in some cases the effects of these structures are not perfectly understood.

When radar data are used in a distributed model, the highest available rainfall time resolution does not always provide the

best estimation of peak flow (Atencia et al., 2011). Under the assumption of a perfect hydrological model with a perfect rainfall input, the accuracy of model output should increase with increase of resolution of both model and rainfall input. In reality, there are some limitations, due to uncertainty and errors related to rainfall measurements and model characteristics (Rafieeinasab et al., 2015).

Many studies present hydrological models, with different characteristics and different representations of the catchment

spatial variability. These models have become more and more detailed, reaching high levels of spatial resolution. If there are no high resolution rainfall data, high model resolution is not the best choice. Rainfall measurement and hydrological modelling resolutions need to be in agreement. Increasing availability of higher resolution rainfall data from rainfall radars will enable deeper investigation of this relationship.

Improved rainfall measurements have also allowed to investigate the relations between temporal and spatial rainfall scale.

Relations have been presented, mostly adapting the Kolgomorov's theory to rainfall, to define the interaction between spatial and temporal scale in atmosphere. A unique relationship has not been find yet, and further investigations are necessary in this direction.

## 8   Conclusions

The relevance of the spatial and temporal variability of rainfall, hydrological processes and catchment characteristics and the

impact that they have on the hydrological response have been discussed in this paper. The main key points and conclusion of this study are the following.

- Hydrological phenomena and processes have different spatial and temporal variability, and sometimes instruments are not able to measure the considered process at the relevant scale. It is important to study hydrological problems at a spatial and temporal scale that is in agreement with the variability of the processes involved.

- Uncertainty associated with rainfall spatial and temporal variability is one of the main sources of error in the estimation of hydrological response in urban areas. New technologies have been developed to measure rainfall spatial and temporal variability more accurately and at higher resolution. While rain gauges remain the most common used rainfall



measurement instruments, weather radars are increasingly used to measure rainfall distributed in space. Especially in urban areas, rain gauges present many limitations due to strong microclimatic variability, complicating identification of suitable locations for representative rainfall measurements. Additionally, rain gauges provide a poor representation of spatial variability because they measure rainfall only at a specific point.

– Infiltration, local storage, interception and evaporation are quite difficult to measure, especially in urban areas, because of the strong heterogeneity of urban land-use.

  – Different types of hydrological models have been developed in order to represent the spatial variability of catchment properties, such as land cover and imperviousness degree. Models can be classified based on their ability to represent the spatial variability of the catchment into lumped, semi-distributed and fully distributed models. Fully distributed models

represent catchments in a more detailed way, but this also requires a higher resolution of the rainfall used as input for the higher model resolution to be beneficial.

  – The impact of spatial and temporal rainfall variability on the hydrological response in urban areas and the role of drainage infrastructure and man-made control structures herein still remains poorly understood. It was found that sensitivity of hydrological response to spatial and temporal rainfall variability varies with catchment size, catchment shape, storm

scale and storm velocity. So far, findings are mainly based on sensitivity studies using theoretical model scenarios. A wider range of conditions and scenarios based on observational datasets for urban hydrological basins need to be analysed in order to characterize better the hydrological response and its sensitivity to different spatial and temporal rainfall resolutions.

*Acknowledgements.* This work has been funded by the EU INTERREG IVB RainGain Project. The authors would like to thank the RainGain

Project (www.raingain.eu) for supporting this research.



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

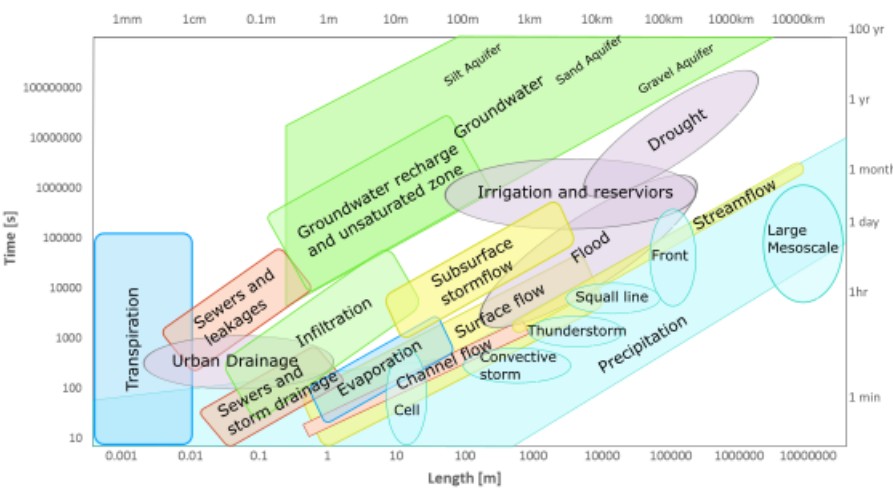

**Figure 1.** Spatial and temporal scale varibility of hydrological processes, adapted from Berndtsson and Niemczynowicz (1986), Blöschl and Sivapalan (1995), Stahl and Hisdal (2004) and Salvadore et al. (2015). Colours represent different groups of physical processes: blue for processes related to the atmosphere, yellow for surface processes, green for underground processes, red highlights typical urban processes and grey indicates problems hydrological processes can pose to society.





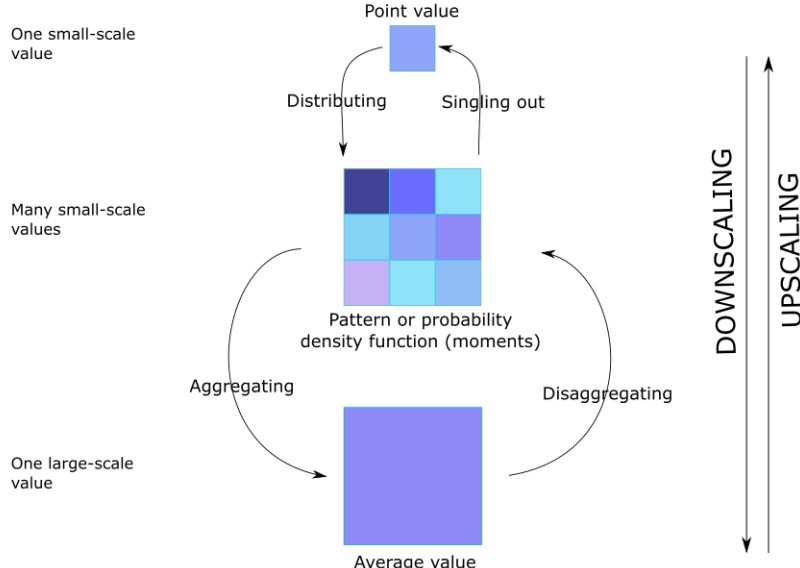

**Figure 2.** Downscaling and upscaling processes (modified from Blöschl and Sivapalan (1995))





**Table 1.** Time scale parameters

| Characteristic | Reference | Description |
|---|---|---|
| Time of concentration $t_c$ | Singh (1997) Gericke and Smithers (2014) | The time that a drop that falls on the most remote part of the drainage basin needs to reach the basin outlet |
| Time of equilibrium $t_e$ | Ogden et al. (1995) Ogden and Dawdy (2003) van de Giesen et al. (2005) | Minimum time needed for a given stationary uniform rainfall to persist until equilibrium runoff flow is reached |
| Lag time $t_{lag}$ | Berne et al. (2004) Marchi et al. (2010) Gericke and Smithers (2014) | The time difference between the gravity center of the hyetograph of catchment mean rainfall and the gravity center of the generated hydrograph |
| Response time scale | Morin et al. (2001) Morin et al. (2002) Morin et al. (2003) Shamir et al. (2005) | The time scale at which the pattern of time averaged radar hyetograph is most similar to the pattern of the measured hydrograph at the outlet of the basin |





**Table 2.** Characterization of rainfall events, spatial and temporal scales and rainfall estimation uncertainty. From van de Beek et al. (2010); Smith et al. (1994) and Emmanuel et al. (2012)

| | Characterization and Intensity | Spatial Range | Temporal Range | Radar Estimation |
|---|---|---|---|---|
| Light rainfall | $1mmh^{-1}$ | $17km$ | $15min$ | Underestimation rainfall values often below the threshold ($0.17mmh^{-1}$) |
| Convective Cells | short and intense from $25mm$ | $5km$ | $5min$ | Overestimation |
| Organized Stratiform | up to $17mmh^{-1}$ | $< 5km$ | $< 5min$ | General underestimation, good representation of the hyetograph behaviour |
| Unorganized Stratiform | intense peak inside intensity rainfall lower | $15km$ | $15min$ | Underestimation of the peaks, good representation of the hyetograph behaviour |





**Table 3.** Critical Resolutions in relation with the drainage area

| Drainage Area DA (ha) | Critical spatial resolution (m$x$m) | Critical temporal resolution (min) |
|---|---|---|
| DA<1 | 100 | 1 |
| 1<DA<100 | 500 | 1 |
| 250<DA<900 | 1000 | <5 |