# Peer review of "Spatial and temporal variability of rainfall and their effects on hydrological response in urban areas - a review"

_Hydrology and Earth System Sciences, 2016_

## Referee Comment (RC1) · Anonymous Referee #1 · 5 Nov 2016

**General**

The manuscript entitled "Spatial and temporal variability of rainfall and their effects on hydrological response in urban areas - a review" by Cristiano et al. provides a literature review of the current understanding of hydrological processes in urban environments with a focus on spatial and temporal variability and scales. It is well written and understandable and would fit well into the scope of HESS special issue on rainfall and urban hydrology. I found no major issues or concerns to be addressed while reading the manuscript, but I have some suggestions of expending the content discussed in part of the sections. Please find my comments, corrections and suggestions below.

**Specific comments [Page; Lines]**

[1; 19-20] For that you can also add the relatively new use of high-quality imagery from unmanned aerial vehicles (UAVs). See the paper by Tokarczyk et al. (2015).

Tokarczyk, P., Leitao, J. P., Rieckermann, J., Schindler, K., and Blumensaat, F.: High-quality observation of surface imperviousness for urban runoff modelling using UAV imagery, Hydrol. Earth Syst. Sci., 19, 4215-4228, doi:10.5194/hess-19-4215-2015, 2015.

[1; 22] Typo: double bracket.

[1; 24-25] This is more or less a repetition of the last sentence of the previous paragraph.

[2; 5] "many aspects" – such as?

[2; 13-14] "Section 7, main knowledge gaps are identified for the with respect to accurate prediction" – please revise.

[2; 23] The colon is not needed.

[3; 3] The title is not completely accurate as you specifically refer to downscaling and upscaling of climate variable to be used as input into hydrological models.

[3; 9] "meteohydrology" – I believe that the term "hydrometeorology" is more common.

[3; 13] "Muthusamy et al., 2016" – this paper discuss upscaling rather than downscaling. I suggested changing the beginning of the sentence to "Statistical downscaling and upscaling approaches … ".

[3; 14] In addition to the paper by Wilby and Wigley (1997) I would also suggest the authors to add the review paper of Wilks and Wilby from 1999 and the (relatively newer) paper by Fowler et al (2007). After all, this is a review paper that should cover the benchmarked papers in the field.

Fowler, H. J., Blenkinsop, S., and Tebaldi, C.: Linking climate change modelling to impacts studies: recent advances in downscaling techniques for hydrological modelling, International Journal of Climatology, 27, 1547–1578, doi:10.1002/joc.1556, 2007.

Wilks, D. S. and Wilby, R. L.: The weather generation game: a review of stochastic weather models, Progress in Physical Geography, 23, 329–357, 1999.

[3; 18-19] I believe that some progress has been made in the AR methods that are being used to generate distributed rainfall since the papers of Ferraris (2003) and Schertzer and Lovejoy (2011). I would suggest the authors to modify the sentence in lines 18-19 to account for some of the relatively new publications in the field. Maybe something like: "Autoregressive methods, also refer to nowadays as "rainfall generator models", are used to generate multidimensional random fields while preserving the rainfall spatial autocorrelation (e.g. Paschalis et al., 2013; Peleg and Morin, 2014; Niemi et al., 2016)".

The three references represent the state of the art high resolution rainfall generator that are now available: STREAP (Paschalis), HiReS-WG (Peleg) and STEPS (Niemi). To that you can probably add the paper by McRobie et al. (2013) in which they extended the earlier Willems model to generate spatially distributed Gaussian rainfall cells (alternatively, this can go to the last model type you are suggesting in this paragraph).

Paschalis, A., Molnar, P., Fatichi, S., and Burlando, P.: A stochastic model for high-resolution space-time precipitation simulation, Water Resources Research, 49, 8400–8417, doi:10.1002/2013WR014437, http://dx.doi.org/10.1002/2013WR014437, 2013.

Peleg, N. and Morin, E.: Stochastic convective rain-field simulation using a high-resolution synoptically conditioned weather generator (HiReS-WG),Water Resources Research, 50, 2124–2139, doi:10.1002/2013WR014836, http://dx.doi.org/10.1002/2013WR014836, 2014.

McRobie, F. H., Wang, L.-P., Onof, C., and Kenney, S.: A spatial-temporal rainfall generator for urban drainage design, Water Science and Technology, 68, 240–249, doi:10.2166/wst.2013.241, 2013.

Niemi, T. J., Guillaume, J. H. A., Kokkonen, T., 5 Hoang, T. M. T., and Seed, A. W.: Role of spatial anisotropy in design storm generation: Experiment and interpretation, Water Resources Research, 52, 69–89, doi:10.1002/2015WR017521, http://dx.doi.org/10.1002/2015WR017521, 2016.

[4; 8-10] There is a repetition with the previous sentence.

[4; 16] Typo: double bracket.

[4; 30-31] "… as, for example the approximation presented by Gericke and Smithers (2014), for which tlag = 0.6tc" – consider deleting this sentence, I don't see how this example can contribute to the reader.

[5; 9] "… the behaviour of four Israelian catchments" – instead: "… the behaviour of four rural catchments in Israel".

[5; 20] Tab. -> Table. Also when tables 2 and 3 are mentioned.

[5; 24] Typo: double "and".

[6; 3] You are referring to five papers as an example to "urban catchments" while not referring at all to studies on "natural watershed", although there are plenty of papers to choose from. I would suggest adding 2-3 references to benchmark papers discussing the use of weather radar in rural catchments as well.

If already mentioning papers that are related to radar and urban hydrology, there is also the paper by Thorndahl et al (2016) which is part of the special issue and I think should also be mention in this review paper.

Thorndahl, S., Einfalt, T., Willems, P., Nielsen, J. E., ten Veldhuis, M.-C., Arnbjerg-Nielsen, K., Rasmussen, M. R., and Molnar, P.: Weather radar rainfall data in urban hydrology, Hydrol. Earth Syst. Sci. Discuss., doi:10.5194/hess-2016-517, in review, 2016.

[6; 8-9] Consider also to add to this references the paper by Fencl et al. (2016), which is going to be published as part of this special issue.

Fencl, M., Dohnal, M., Rieckermann, J., and Bareš, V.: Gauge-Adjusted Rainfall Estimates from Commercial Microwave Links, Hydrol. Earth Syst. Sci. Discuss., doi:10.5194/hess-2016-397, in review, 2016.

[6; 20-21] "To solve the problem of spatial representation, interpolation techniques are used to obtain distributed rainfall fields…" – good. But sometimes you wish to do the opposite, go from a data obtained by a dense rain—gauge network to the areal rainfall that represents the catchment. This is the upscaling paper by Muthusamy et al. (2016) that was mentioned on [3; 13].

[6; 25-26] "A second problem is introduced by hard surfaces, that may cause water splashing into the gauges" – I thought that the recommendation of the WMO are to mount the gauges at an elevation of 1.2 m above ground. If this is the case, I don't think water splashing is an issue.

[6; 32] "F. and A." – please correct.

[6; 33] Typo: double bracket.

[7; 6-7] Consider presenting the comparison between the different band widths in a table. Maybe add price estimation for each radar type?

[7; 14] Limitations of rain—gauges are not discussed in this section. Please remove "abd rain gauges" from the section title.

[7; 19-20] "…and to define the uncertainty related to radar-rainfall estimation (Mandapaka et al., 2009; Overeem et al., 2009a)" – I suggest the authors to remove the reference to Overeem from this sentence (but to keep this reference when it cited next in the paragraph) and to replace it with other studies that were more focusing on rainfall-radar uncertainties, such as: Ciach and Krajewski (1999), Villarini et al. (2008) and Peleg et al. (2013).

Ciach, G. J. and Krajewski, W. F.: On the estimation of radar rainfall error variance, Adv. Water Resour., 22, 585–595, doi:10.1016/s0309-1708(98)00043-8, 1999.

Villarini, G., Mandapaka, P. V., Krajewski, W. F., and Moore, R. J.: Rainfall and sampling uncertainties: A rain gauge perspective, J. Geophys. Res.-Atmos., 113, D11102, doi:10.1029/2007jd009214, 2008.

Peleg, N., Ben-Asher, M., and Morin, E.: Radar subpixel-scale rainfall variability and uncertainty: lessons learned from observations of a dense rain-gauge network, Hydrology and Earth System Sciences, 17, 2195–2208, doi:10.5194/hess-17-2195-2013, 2013.

[7; 32-33] A repetitive sentence. Consider deleting.

[8; 7-8] I think that an operative rainfall forecast based on weather radar has been activated in Belgium (using STEP model). Please check the following paper:

Foresti, L., M. Reyniers, A. Seed, and L. Delobbe. "Development and verification of a real-time stochastic precipitation nowcasting system for urban hydrology in Belgium." Hydrology and Earth System Sciences 20 (2016): 505-527.

[8; 10-12] Pollution due to urbanization also affects rainfall. Check for example the paper mention below. I would also suggest add to modify the sentence in line 11 accordingly: "Increase in heat and pollution produced by human activities …".

Givati, A., & Rosenfeld, D. (2004). Quantifying precipitation suppression due to air pollution. Journal of Applied meteorology, 43(7), 1038-1056.

[8; 25] Urban areas? You have a word missing.

[9; 6-23] This read to me as a separate subsection, entitled as "rainfall variability at the urban scale", or alike.

[9; 16-17] Not necessarily, the setup needed for deployment of a dense rain-gauge network at the urban scale that can well represent the rainfall spatial variability can be calculated using the variance reduction factor. See papers by Villarini et al. (2008) and Peleg et al. (2013) that were suggested above.

[9; 23] Please also have a look at the recent paper by Peleg et al., whom examined the spatial distribution of extreme rainfall intensity for the same scale and using similar methods (but with different rainfall model) as Gires et al. mentioned here. They found that the spatial distribution of extreme rainfall over small domains (1 x 1 km$^2$) can be very high.

Peleg, N., Marra, F., Fatichi, S., Paschalis, A., Molnar, P., and Burlando, P.: Spatial variability of extreme rainfall at radar subpixel scale, Journal of Hydrology, doi:doi:10.1016/j.jhydrol.2016.05.033, 2016.

[9; 27] What do you mean by common? C-band radars?

[10; 27-28] Please delete. It repeats what is already mention.

[10; 29] Consider changing the title to: "Groundwater recharge and subsurface processes in urban areas". Infiltration is already discussed in the previous paragraph.

[11; 10-12] Please revise this sentence.

[12; 22] "Fr" in italic.

[13; 20] A reference is needed here.

[13; 21] Delete "Recent", a study from 1991 cannot be consider as recent…

[13; 23] Typo: delete the commas.

[13; 30] Why the catchments needs not to be placed on concrete or asphalt?

[15; 2-7] Please give references and examples to the two types of models used in urban studies.

[16; 14-18] Please indicate full names for UDTM and EPA SWMM models. A reference for the SWMM model should also be added.

[16; 2-24] It can be useful to add a table with the most common hydrodynamic models that are been used in urban studies (including full name, abbreviation, reference and the model type).

[17; 12] There is another relevant paper that is a part of this special issue (see below). It deals with the effect of spatially distributed rainfall on the flow total variability in an urban catchment.

Peleg, N., Blumensaat, F., Molnar, P., Fatichi, S., and Burlando, P.: Partitioning spatial and temporal rainfall variability in urban drainage modelling, Hydrol. Earth Syst. Sci. Discuss., doi:10.5194/hess-2016-530, in review, 2016.

[17; 30] Should be "authors".

Figure 2 – Consider changing the pixel to a point for the point value (or add a point within the pixel).

Table 1 – Notatains as missing in the box of the Response time scale.

Table 3 – Correct to "(m x m)".

---

## Author Comment (AC1) · 13 Dec 2016

RC = Reviewer comment AR = Authors reply

General:

RC: The manuscript entitled "Spatial and temporal variability of rainfall and their effects on hydrological response in urban areas - a review" by Cristiano et al. provides a literature review of the current understanding of hydrological processes in urban environments with a focus on spatial and temporal variability and scales. It is well written and understandable and would fit well into the scope of HESS special issue on rainfall and urban hydrology. I found no major issues or concerns to be addressed while reading the manuscript, but I have some suggestions of expending the content discussed in part of the sections. Please find my comments, corrections and suggestions below.

AC: On behalf of the authors, I would like to thank the reviewer for the effort and time spend to analyse this manuscript. We greatly appreciate the reviewer's comments and suggestions that will help to improve the quality of the manuscript.

Specific comments [Page; Lines]

RC: [1; 22] Typo: double bracket.

[2; 5] "many aspects" – such as?

[2; 13-14] "Section 7, main knowledge gaps are identified for the with respect to accurate prediction" – please revise.

[2; 23] The colon is not needed.

[3; 3] The title is not completely accurate as you specifically refer to downscaling and upscaling of climate variable to be used as input into hydrological models.

[3; 9] "meteohydrology" – I believe that the term "hydrometeorology" is more common.

[3; 13] "Muthusamy et al., 2016" – this paper discuss upscaling rather than downscaling. I suggested changing the beginning of the sentence to "Statistical downscaling and upscaling approaches . . . ".

[4; 16] Typo: double bracket.

[5; 9] ". . . the behaviour of four Israelian catchments" – instead: ". . . the behaviour of four rural catchments in Israel".

[5; 20] Tab. -> Table. Also when tables 2 and 3 are mentioned.

[5; 24] Typo: double "and".

[6; 32] "F. and A." – please correct.

[6; 33] Typo: double bracket.

[7; 14] Limitations of rain—gauges are not discussed in this section. Please remove "abd rain gauges" from the section title.

[8; 25] Urban areas? You have a word missing.

[9; 6-23] This read to me as a separate subsection, entitled as "rainfall variability at the urban scale", or alike.

[9; 27] What do you mean by common? C-band radars?

[10; 29] Consider changing the title to: "Groundwater recharge and subsurface processes in urban areas". Infiltration is already discussed in the previous paragraph.

[11; 10-12] Please revise this sentence.

[12; 22] "Fr" in italic.

[13; 20] A reference is needed here.

[13; 21] Delete "Recent", a study from 1991 cannot be consider as recent. . .

[13; 23] Typo: delete the commas.

[13; 30] Why the catchments needs not to be placed on concrete or asphalt?

[17; 30] Should be "authors".

Table 1 – Notatains as missing in the box of the Response time scale.

Table 3 – Correct to "(m x m)".

AC: Thank you for catching the typo mistakes. They will be corrected. Vague sentences will be revised and proper references will be added where necessary.

RC: [1; 19-20] For that you can also add the relatively new use of high-quality imagery from unmanned aerial vehicles (UAVs). See the paper by Tokarczyk et al. (2015).

Tokarczyk, P., Leitao, J. P., Rieckermann, J., Schindler, K., and Blumensaat, F.: High-quality observation of surface imperviousness for urban runoff modelling using UAV imagery, Hydrol. Earth Syst. Sci., 19, 4215-4228, doi:10.5194/hess-19-4215-2015, 2015.

[3; 14] In addition to the paper by Wilby and Wigley (1997) I would also suggest the authors to add the review paper of Wilks and Wilby from 1999 and the (relatively newer) paper by Fowler et al (2007). After all, this is a review paper that should cover the benchmarked papers in the field.

Fowler, H. J., Blenkinsop, S., and Tebaldi, C.: Linking climate change modelling to impacts studies: recent advances in downscaling techniques for hydrological modelling, International Journal of Climatology, 27, 1547–1578, doi:10.1002/joc.1556, 2007.

Wilks, D. S. and Wilby, R. L.: The weather generation game: a review of stochastic weather models, Progress in Physical Geography, 23, 329–357, 1999. [3; 18-19] I believe that some progress has been made in the AR methods that are being used to generate distributed rainfall since the papers of Ferraris (2003) and Schertzer and Lovejoy (2011).

I would suggest the authors to modify the sentence in lines 18-19 to account for some of the relatively new publications in the field. Maybe something like: "Autoregressive methods, also refer to nowadays as "rainfall generator models", are used to generate multidimensional random fields while preserving the rainfall spatial autocorrelation (e.g. Paschalis et al., 2013; Peleg and Morin, 2014; Niemi et al., 2016)". The three references represent the state of the art high resolution rainfall generator that are now available: STREAP (Paschalis), HiReS-WG (Peleg) and STEPS (Niemi). To that you can probably add the paper by McRobie et al. (2013) in which they extended the earlier Willems model to generate spatially distributed Gaussian rainfall cells (alternatively, this can go to the last model type you are suggesting in this paragraph).

Paschalis, A., Molnar, P., Fatichi, S., and Burlando, P.: A stochastic model for

high resolution space-time precipitation simulation, Water Resources Research, 49, 8400– 8417, doi:10.1002/2013WR014437, http://dx.doi.org/10.1002/2013WR014437, 2013.     Peleg, N. and Morin, E.:   Stochastic convective rain-field simulation using a high-resolution synoptically conditioned weather generator (HiReS-WG),Water Resources Research, 50, 2124–2139, doi:10.1002/2013WR014836, http://dx.doi.org/10.1002/2013WR014836, 2014. McRobie, F. H., Wang, L.-P., Onof, C., and Kenney, S.:  A spatial-temporal rainfall generator for urban drainage design, Water Science and Technology, 68, 240–249, doi:10.2166/wst.2013.241, 2013. Niemi, T. J., Guillaume, J. H. A., Kokkonen, T., 5 Hoang, T. M. T., and Seed, A. W.:  Role of spatial anisotropy in design storm generation: Experiment and interpretation, Water Resources Research, 52, 69–89, doi:10.1002/2015WR017521, http://dx.doi.org/10.1002/2015WR017521, 2016.

[6; 3] You are referring to five papers as an example to "urban catchments" while not referring at all to studies on "natural watershed", although there are plenty of papers to choose from. I would suggest adding 2-3 references to benchmark papers discussing the use of weather radar in rural catchments as well.  If already mentioning papers that are related to radar and urban hydrology, there is also the paper by Thorndahl et al (2016) which is part of the special issue and I think should also be mention in this review paper.

Thorndahl, S., Einfalt, T., Willems, P., Nielsen, J. E., ten Veldhuis, M.-C., Arnbjerg-Nielsen, K., Rasmussen, M. R., and Molnar, P.: Weather radar rainfall data in urban hydrology, Hydrol. Earth Syst. Sci. Discuss., doi:10.5194/hess-2016-517, in review, 2016.

[6; 8-9] Consider also to add to this references the paper by Fencl et al. (2016), which is going to be published as part of this special issue. Fencl, M., Dohnal, M., Rieckermann, J., and Bareš, V.: Gauge-Adjusted Rainfall Estimates from Commercial Microwave Links, Hydrol. Earth Syst. Sci. Discuss., doi:10.5194/hess- 2016-397, in review, 2016.

[Figure]

[6; 20-21] "To solve the problem of spatial representation, interpolation techniques are used to obtain distributed rainfall fields..." – good. But sometimes you wish to do the opposite, go from a data obtained by a dense rain—gauge network to the areal rainfall that represents the catchment. This is the upscaling paper by Muthusamy et al. (2016) that was mentioned on [3; 13].

[7; 19-20] "...and to define the uncertainty related to radar-rainfall estimation (Mandapaka et al., 2009; Overeem et al., 2009a)" – I suggest the authors to remove the reference to Overeem from this sentence (but to keep this reference when it cited next in the paragraph) and to replace it with other studies that were more focusing on rainfall radar uncertainties, such as: Ciach and Krajewski (1999), Villarini et al. (2008) and Peleg et al. (2013).

Ciach, G. J. and Krajewski, W. F.: On the estimation of radar rainfall error variance, Adv. Water Resour., 22, 585–595, doi:10.1016/s0309-1708(98)00043-8, 1999.

Villarini, G., Mandapaka, P. V., Krajewski, W. F., and Moore, R. J.: Rainfall and sampling uncertainties: A rain gauge perspective, J. Geophys. Res.-Atmos., 113, D11102, doi:10.1029/2007jd009214, 2008.

Peleg, N., Ben-Asher, M., and Morin, E.: Radar subpixel-scale rainfall variability and uncertainty: lessons learned from observations of a dense rain-gauge network, Hydrology and Earth System Sciences, 17, 2195–2208, doi:10.5194/hess-17-2195-2013, 2013.

[8; 7-8] I think that an operative rainfall forecast based on weather radar has been activated in Belgium (using STEP model). Please check the following paper:

Foresti, L., M. Reyniers, A. Seed, and L. Delobbe. "Development and verification of a realtime stochastic precipitation nowcasting system for urban hydrology in Belgium." Hydrology and Earth System Sciences 20 (2016): 505-527.

[8; 10-12] Pollution due to urbanization also affects rainfall. Check for example the

paper mention below. I would also suggest add to modify the sentence in line 11 accordingly: "Increase in heat and pollution produced by human activities . . .". Givati, A., & Rosenfeld, D. (2004). Quantifying precipitation suppression due to air pollution. Journal of Applied meteorology, 43(7), 1038-1056.

[9; 16-17] Not necessarily, the setup needed for deployment of a dense rain-gauge network at the urban scale that can well represent the rainfall spatial variability can be calculated using the variance reduction factor. See papers by Villarini et al. (2008) and Peleg et al. (2013) that were suggested above.

[9; 23] Please also have a look at the recent paper by Peleg et al., whom examined the spatial distribution of extreme rainfall intensity for the same scale and using similar methods (but with different rainfall model) as Gires et al. mentioned here. They found that the spatial distribution of extreme rainfall over small domains (1 x 1 km2 ) can be very high.

Peleg, N., Marra, F., Fatichi, S., Paschalis, A., Molnar, P., and Burlando, P.: Spatial variability of extreme rainfall at radar subpixel scale, Journal of Hydrology, doi:doi:10.1016/j.jhydrol.2016.05.033, 2016.

[17; 12] There is another relevant paper that is a part of this special issue (see below). It deals with the effect of spatially distributed rainfall on the flow total variability in an urban catchment. Peleg, N., Blumensaat, F., Molnar, P., Fatichi, S., and Burlando, P.: Partitioning spatial and temporal rainfall variability in urban drainage modelling, Hydrol. Earth Syst. Sci. Discuss., doi:10.5194/hess-2016-530, in review, 2016.

AC: Thank you for these great suggestions. Recommended papers will be added

RC: [1; 24-25] This is more or less a repetition of the last sentence of the previous paragraph

[4; 8-10] There is a repetition with the previous sentence.

[4; 30-31] ". . . as, for example the approximation presented by Gericke and Smithers

(2014), for which tlag = 0.6tc" – consider deleting this sentence, I don't see how this example can contribute to the reader.

[7; 32-33] A repetitive sentence. Consider deleting.

[10; 27-28] Please delete. It repeats what is already mention.

[6; 25-26] "A second problem is introduced by hard surfaces, that may cause water splashing into the gauges" – I thought that the recommendation of the WMO are to mount the gauges at an elevation of 1.2 m above ground. If this is the case, I don't think water splashing is an issue.

AC: Thanks for the suggestions. Repetitive and not necessary sentence will be deleted.

RC: [15; 2-7] Please give references and examples to the two types of models used in urban studies.

[16; 14-18] Please indicate full names for UDTM and EPA SWMM models. A reference for the SWMM model should also be added.

[16; 2-24] It can be useful to add a table with the most common hydrodynamic models that are been used in urban studies (including full name, abbreviation, reference and the model type).

AC: Thanks for valuable suggestions. The authors agreed that the section related to urban hydrodynamic models can be extended adding more references and a summary table of the most common model used in urban studies.

RC: [7; 6-7] Consider presenting the comparison between the different band widths in a table. Maybe add price estimation for each radar type?

AC: We appreciate the reviewer's suggestion. A comparison between different radar types will be presented in a table, that summarize the radar characteristics.

RC: Figure 2 – Consider changing the pixel to a point for the point value (or add a point

within the pixel).

AC: Thanks for the suggestions. Figure 2 will be fixed adding a point within the pixel.
* * *

---

## Referee Comment (RC2) · Anonymous Referee #2 · 21 Jan 2017

Review comments Spatial and temporal variability of rainfall and their effects on hydrological response in urban areas - a review by Elena Cristian, Marie-Claire ten Veldhius, and Nick van de Giesen

General comments: This manuscript is a scientific review on the variability of rainfall in urban hydrology. Several review papers have recently been published on urban hydrology in the main journals. An additional one could be interesting if it proposes an original point of view. As far as I know, it seems to be the case for this manuscript. Nevertheless, I consider that the manuscript requires a significant revision before its publication in HESS can be considered. - The manuscript covers a wide range of topics, sometimes from a very general point of view (for instance disaggregation, but not only), and

sometimes already very well documented in text books (for instance hydrological processes). It is not consistent with a review paper which should present the state of the art of research on the addressed subject. The authors are recommended to focus on the most original part of the manuscript, for which a review presents an added value. - I have in mind several papers, which address the subject of the manuscript, and which have been omitted. I recommend that the authors provide a more comprehensive and exhaustive state of the art, representative of the recent studies. In addition, the references to studies of urban basins (instead of, or in addition to, natural ones) would be welcome in a manuscript covering urban areas. Specific comments

p1. 15-18: This sentence is questionable. Hydrologists have been working on rainfall radar measurement for a very long time, and have significantly contributed to this subject, including in urban hydrology (Einfalt et al., 2004 for a review). The cited references are recent and don't reflect this long term research effort which has known a renewed interest with the emergence of polarimetric X-band radars which allow to solve some problems met with classic low-cost radars. p2. 1-3: I am not sure that it is more complicated, I would say different. p3.1-2 : I don't well understand this sentence p3. 3-21: Downscaling and upscaling in hydrology: This paragraph is a very brief and general introduction of downscaling methods. It is not very useful for the reader because the authors don't refer to the applications of these methods in urban hydrology, which is the subject of the manuscript. p3. 27-30: the paper by Julien and Moglen (1990) doesn't address the particular case of urban catchments. p4. 6-11: I have read the paper by Gericke and Smithers (2014). Their review of the existing methods doesn't address the urban basins. p4. 13-14: The production function is very different in natural areas and urban areas. Initial losses don't exceed 1 or 2 mm, and most of the impervious surfaces are directly connected to the hydrographic (or sewer) network. This statement is not valid for urban basins. p4. 24: The term response time is also often used; this term may be similar to the response time (Musy, 2011). p4. 32-33: To the best of my knowledge, the paper by Morin et al. (2001) doesn't address urban basins paragraph 2.3.2: "time scale characteristics". It is interesting to highlight these

terms which characterize the basin dynamics are less used nowadays (see the topic called synthetic hydrology which addresses this subject). In my opinion, there are more or less equivalent, and too many words are used to name very close concepts, which can be confusing. It could be the opportunity to propose one or two terms. I think that most of the papers that are cited (not very recent) don't concern urban basins, which is a problem concerning the subject of this manuscript. I know that similar relations have been proposed (at the same period) in urban hydrology to relate the response (or lag) time of urban basins to the characteristics of basins : surface, imperviousness, slope, roughness . . . I recommend to add references on time scale characteristics that address urban basins. p6. 15-20: I suggest to refer to Lanza and Stagi (2009) and or to Lanza and Vuerich (2009) for a recent evaluation lab. and field evaluation and comparison of rain gauges. p6. 18-19: the rain gauge data is punctual, but as rainfall field displays a spatial organization, this data is representative of rainfall in its neighborhood, in a surface area which depends on time step and decorrelation distance, itself related to the rainfall type. p6. 21: It exists in the scientific literature many papers, including review papers, dealing with the interpolation of rain gauges data for mapping rainfall fields by various methods, and I don't understand why the authors refer to Shaghaghian and Abedini (2013) not published in an international journal. p7. 10-14: The added value of polarimetric data is mainly: i) for ground clutter detection removal, and for X-band (and C-band also) radars for attenuation correction. X-band are strongly affected by attenuation of the signal by rainfall, and the correction of this problem is very unstable. Polarimetric data allow to efficiently correct this problem. p7. 15-25 : I don't agree with this paragraph for two reasons. The authors refer to a few studies dealing with the calibration of radar data by rain gauges, or the combination of radar and rain gauges data (which is a much more recent approach of this question). I think that these studies are not representative of the state of knowledge on that subject. In addition, based on a very limited number of studies, they conclude to the underestimation of rainfall by radar. I consider that this conclusion is erroneous and not justified p7. 28: Be more precise, please! The radar equation relates the backscattered power

to the radar reflectivity factor, usually called reflectivity. This radar equation doesn't depend on the drop size distribution. p8 : paragraph 3.2 "Influence of urban areas on rainfall". This paragraph provides a brief, and general overview of this subject. What is its interaction with the subject of the manuscript ? Is this paragraph useful ? I am not convinced. p8-9. Paragraph 3.3. It seems that the weather is a very convenient device to analyze the spatial and temporal scales of rain fields for urban hydrology. p9 to p14: section 4 Hydrological processes. This very long paragraph summarizes the main hydrological processes and the main approaches used to represent them, in natural and urban areas as well. It regroups the basic knowledge in hydrology, addressed in text books, and not suited to a review manuscript. This section must be removed. p15. 1-6: I don't understand these sentences. Are stochastic models used in urban hydrology ? What type of models for what application? I don't see its usefulness in this manuscript. It could be confusing for the reader. Please, remove it! p15. 11-14: As I understand this criterion, it concerns only the influence of the spatial variability of rainfall. There is many other factors involved in the choice of an hydrological model, depending on the applications of this model. p15. 24-31. The notion of physically-based or conceptual is valid at a given scale (in my opinion). The computational power is no more a problem, but I agree that the parameterization of a model, and the values to assign to these parameters remains a key issue. p16. 2-3 : it is an important element for all models at all scales p16. 5-13: I agree p16. 14-25: Meselhe et al. (2008) : I have not found this article in WRR. This example is certainly interesting. Unfortunately, it doesn't deal with urban hydrology. In a review manuscript addressing the hydrological response of urban basins, it is highly recommended to refer to papers which address the subject of the manuscript. p17. 1-12: I would say that this subject is yet an open research subject, and the influence of spatial variability of rainfall on the basin response at its outlet is not yet well understood. A basin is a very powerful filter in time and space which smooths significantly an input impulse. Some studies, mostly dealing with flash floods, have been performed to determine the characteristics of rainfall which explain the variations of hydrographs at the basin outlet and reach different conclusions. p17. 13-16: It is

interesting to keep in mind that a basin is a geographic system, not only characterized by its outlet. A distributed model allows to determine the flow at any location within the basin, if the rainfall is measured at corresponding scales. p17. 18.32 (paragraph 6.1.1): this reasoning applies to calculate the flow at the basin outlet, and it is no more valid if to get the flow at locations within the basin. p17. 25-26: I would suggest be very careful with these relations, which remain only indicative, and subject to a large mean error. For instance, it is very different from the equation 2. I would suggest to keep a critical and consistent approach along the manuscript. p17-18 (paragraph 6.1.2): I suggest to regroup it with 6.1.1. both deal the influence of rainfall variability according to the basin features : surface, length, response time . . .. p17-18. The subject of these two paragraphs (to be regrouped) - influence of spatial and temporal rainfall variability in relation with basin characteristics – is very important, and has been addressed by a large number of papers. The authors present in detail a limited number of studies (five or six). I suggest that they enrich their bibliography on that subject in order to provide a more comprehensive state of the art on that subject.

Lanza LG, Stagi L., 2009. High resolution performance of catching type rain gauges from the laboratory phase of the WMO Field Intercomparison of Rain Intensity Gauges. Atmos Res., 94, 555-563. Lanza L.G., Vuerich E., 2009. The WMO Field Intercomparison of Rain Intensity Gauges. Atmos. Res., 94,4, 534-543.
* * *

---

## Author Comment (AC2) · 29 Jan 2017

RC = Reviewer comment AR = Authors reply

General Comment

RC: Spatial and temporal variability of rainfall and their effects on hydrological response in urban areas - a review by Elena Cristiano, Marie-Claire ten Veldhuis, and Nick van de Giesen

General comments: This manuscript is a scientific review on the variability of rainfall in urban hydrology. Several review papers have recently been published on urban hydrology in the main journals. An additional one could be interesting if it proposes an

original point of view. As far as I know, it seems to be the case for this manuscript. Nevertheless, I consider that the manuscript requires a significant revision before its publication in HESS can be considered. - The manuscript covers a wide range of topics, sometimes from a very general point of view (for instance disaggregation, but not only), and paper sometimes already very well documented in text books (for instance hydrological processes). It is not consistent with a review paper which should present the state of the art of research on the addressed subject. The authors are recommended to focus on the most original part of the manuscript, for which a review presents an added value. - I have in mind several papers, which address the subject of the manuscript, and which have been omitted. I recommend that the authors provide a more comprehensive and exhaustive state of the art, representative of the recent studies. In addition, the references to studies of urban basins (instead of, or in addition to, natural ones) would be welcome in a manuscript covering urban areas.

AR: The authors would like to thank the reviewer for the time and effort spent reviewing our manuscript. We will make sure to check for additional references and to incorporate them in the revised version. The structure of the manuscript will be reconsidered, such that it focuses on the state of the art of research results in an urban hydrology context and addresses natural catchments only in reference to how processes in urban catchments are different. "Already well documented parts" will be removed and reconsidered with the aim of focusing only on recent progress.

Specific comments

RC: p1. 15-18: This sentence is questionable. Hydrologists have been working on rainfall radar measurement for a very long time, and have significantly contributed to this subject, including in urban hydrology (Einfalt et al., 2004 for a review). The cited references are recent and don't reflect this long term research effort which has known a renewed interest with the emergence of polarimetric X-band radars which allow to solve some problems met with classic low-cost radars. p2. 1-3: I am not sure that it is more complicated, I would say different. p3.1-2 : I don't well understand this

sentence p3. 3-21: Downscaling and upscaling in hydrology: This paragraph is a very brief and general introduction of downscaling methods. It is not very useful for the reader because the authors don't refer to the applications of these methods in urban hydrology, which is the subject of the manuscript. p3. 27-30: the paper by Julien and Moglen (1990) doesn't address the particular case of urban catchments. p4. 6-11: I have read the paper by Gericke and Smithers (2014). Their review of the existing methods doesn't address the urban basins. p4. 13-14: The production function is very different in natural areas and urban areas. Initial losses don't exceed 1 or 2 mm, and most of the impervious surfaces are directly connected to the hydrographic (or sewer) network. This statement is not valid for urban basins. p4. 24: The term response time is also often used; this term may be similar to the response time (Musy, 2011). p4. 32-33: To the best of my knowledge, the paper by Morin et al. (2001) doesn't address urban basins paragraph 2.3.2: "time scale characteristics". It is interesting to highlight these terms which characterize the basin dynamics are less used nowadays (see the topic called synthetic hydrology which addresses this subject). In my opinion, there are more or less equivalent, and too many words are used to name very close concepts, which can be confusing. It could be the opportunity to propose one or two terms. I think that most of the papers that are cited (not very recent) don't concern urban basins, which is a problem concerning the subject of this manuscript. I know that similar relations have been proposed (at the same period) in urban hydrology to relate the response (or lag) time of urban basins to the characteristics of basins: surface, imperviousness, slope, roughness . . . I recommend to add references on time scale characteristics that address urban basins.

AR: We agree that the focus of this section should be more on urban hydrology. Parts of this section will be reconsidered and rephrased. The focus will be only on urban areas, presenting results relative to natural catchments only when they are relevant to have a better understanding of the urban environment or if it is interesting to highlight the differences between the two areas. As for the time scales terminology, we have included a summary of terms used in the literature; we're not sure how to interpret the

reviewer's suggestion of proposing a particular term.

RC: p6. 15-20: I suggest to refer to Lanza and Stagi (2009) and or to Lanza and Vuerich (2009) for a recent evaluation lab. and field evaluation and comparison of rain gauges. p6. 18-19: the rain gauge data is punctual, but as rainfall field displays a spatial organization, this data is representative of rainfall in its neighborhood, in a surface area which depends on time step and decorrelation distance, itself related to the rainfall type. p6. 21: It exists in the scientific literature many papers, including review papers, dealing with the interpolation of rain gauges data for mapping rainfall fields by various methods, and I don't understand why the authors refer to Shaghaghian and Abedini (2013) not published in an international journal. p7. 10-14: The added value of polarimetric data is mainly: i) for ground clutter detection removal, and for X-band (and C-band also) radars for attenuation correction. X-band are strongly affected by attenuation of the signal by rainfall, and the correction of this problem is very unstable. Polarimetric data allow to efficiently correct this problem. p7. 15-25 : I don't agree with this paragraph for two reasons. The authors refer to a few studies dealing with the calibration of radar data by rain gauges, or the combination of radar and rain gauges data (which is a much more recent approach of this question). I think that these studies are not representative of the state of knowledge on that subject. In addition, based on a very limited number of studies, they conclude to the underestimation of rainfall by radar. I consider that this conclusion is erroneous and not justified p7. 28: Be more precise, please! The radar equation relates the backscattered power to the radar reflectivity factor, usually called reflectivity. This radar equation doesn't depend on the drop size distribution. p8-9. Paragraph 3.3. It seems that the weather is a very convenient device to analyze the spatial and temporal scales of rain fields for urban hydrology.

AR: Suggested references and others will be added in this section in order to have a better description of the state of the art of rainfall measurements and in particular on the importance of using weather radars combined with rain gauges to estimate rainfall. In particular we are going to refer to Thorndahl et al. 2016, (who offer a good review

on advances in weather radars and their application in urban hydrology ), and to other references that will allow us to give a more precise description of radars, in terms of instrument, measurements and applications. See below for some of the references that will be added.

Thorndahl, S., Einfalt, T., Willems, P., Nielsen, J. E., ten Veldhuis, M.-C., Arnbjerg-Nielsen, K., Rasmussen, M. R., and Molnar, P.: Weather radar rainfall data in urban hydrology, Hydrol. Earth Syst. Sci. Discuss., doi:10.5194/hess-2016-517, in review, 2016.

(on rainfall generator, suggested by reviewer#1)

Paschalis, A., Molnar, P., Fatichi, S., and Burlando, P.: A stochastic model for high resolution space-time precipitation simulation, Water Resources Research, 49, 8400– 8417, doi:10.1002/2013WR014437, http://dx.doi.org/10.1002/2013WR014437, 2013. Peleg, N. and Morin, E.: Stochastic convective rain-field simulation using a high-resolution synoptically conditioned weather generator (HiReS-WG),Water Resources Research, 50, 2124–2139, doi:10.1002/2013WR014836, http://dx.doi.org/10.1002/2013WR014836, 2014.

McRobie, F. H., Wang, L.-P., Onof, C., and Kenney, S.: A spatial-temporal rainfall generator for urban drainage design, Water Science and Technology, 68, 240–249, doi:10.2166/wst.2013.241, 2013.

Niemi, T. J., Guillaume, J. H. A., Kokkonen, T., 5 Hoang, T. M. T., and Seed, A. W.: Role of spatial anisotropy in design storm generation: Experiment and interpretation, Water Resources Research, 52, 69–89, doi:10.1002/2015WR017521, http://dx.doi.org/10.1002/2015WR017521, 2016.

(on rainfall uncertainty suggested by reviewer#1)

Ciach, G. J. and Krajewski, W. F.: On the estimation of radar rainfall error variance, Adv. Water Resour., 22, 585–595, doi:10.1016/s0309-1708(98)00043-8, 1999.

Villarini, G., Mandapaka, P. V., Krajewski, W. F., and Moore, R. J.: Rainfall and sampling uncertainties: A rain gauge perspective, J. Geophys. Res.-Atmos., 113, D11102, doi:10.1029/2007jd009214, 2008.

Peleg, N., Ben-Asher, M., and Morin, E.: Radar subpixel-scale rainfall variability and uncertainty: lessons learned from observations of a dense rain-gauge network, Hydrology and Earth System Sciences, 17, 2195–2208, doi:10.5194/hess-17-2195-2013, 2013.

(on rainfall spatial distribution in 1 radar pixel, suggested by reviewer#1)

Peleg, N., Marra, F., Fatichi, S., Paschalis, A., Molnar, P., and Burlando, P.: Spatial variability of extreme rainfall at radar subpixel scale, Journal of Hydrology, doi:doi:10.1016/j.jhydrol.2016.05.033, 2016.

(other references)

Nielsen, J. E., Thorndahl, S. and Rasmussen, M. R.: Improving weather radar precipitation estimates by combining two types of radars, Atmospheric Research, 139, 36–45, doi:10.1016/j.atmosres.2013.12.013, 2014

Quirmbach, M. and Schultz, G. A.: Comparison of rain gauge and radar data as input to an urban rainfall-runoff model, in Water Science and Technology, vol. 45, pp. 27–33., 2002.

Sørup, H. J. D., Christensen, O. B., Arnbjerg-Nielsen, K. and Mikkelsen, P. S.: Downscaling future precipitation extremes to urban hydrology scales using a spatio-temporal Neyman–Scott weather generator, Hydrology and Earth System Sciences 15 Discussions, 12(2), 2561–2605, doi:10.5194/hessd-12-2561-2015, 2015.

Villarini, G., Seo, B. C., Serinaldi, F. and Krajewski, W. F.: Spatial and temporal modeling of radar rainfall uncertainties, Atmospheric Research, 135–136, 91–101, doi:10.1016/j.atmosres.2013.09.007, 2014.
RC: p8 : paragraph 3.2 "Influence of urban areas on rainfall". This paragraph provides a brief, and general overview of this subject. What is its interaction with the subject of the manuscript ? Is this paragraph useful ? I am not convinced.

AC: The authors agree that the paragraph can be removed. Although it is an important topic, it is not relevant for the scope of the review.

RC: p9 to p14: section 4 Hydrological processes. This very long paragraph summarizes the main hydrological processes and the main approaches used to represent them, in natural and urban areas as well. It regroups the basic knowledge in hydrology, addressed in text books, and not suited to a review manuscript. This section must be removed.

AR: This section will be reconsidered, focusing only on the recent findings that are relevant in urban areas. The section will be largely reduced in order to highlight the main aspects that represent the most recent findings in urban hydrology.

RC: p15. 1-6: I don't understand these sentences. Are stochastic models used in urban hydrology ? What type of models for what application? I don't see its usefulness in this manuscript. It could be confusing for the reader. Please, remove it! p15. 11-14: As I understand this criterion, it concerns only the influence of the spatial variability of rainfall. There is many other factors involved in the choice of an hydrological model, depending on the applications of this model. p15. 24-31. The notion of physically-based or conceptual is valid at a given scale (in my opinion). The computational power is no more a problem, but I agree that the parameterization of a model, and the values to assign to these parameters remains a key issue. p16. 2-3 : it is an important element for all models at all scales p16. 5-13: I agree p16. 14-25: Meselhe et al. (2008): I have not found this article in WRR. This example is certainly interesting. Unfortunately, it doesn't deal with urban hydrology. In a review manuscript addressing the hydrological response of urban basins, it is highly recommended to refer to papers which address the subject of the manuscript.

[Figure]

AR: The model section will be restructured focusing more on the models that are used in urban areas. A table that include a list of most common and used urban hydrodynamic models will be added.

RC: p17. 1-12: I would say that this subject is yet an open research subject, and the influence of spatial variability of rainfall on the basin response at its outlet is not yet well understood. A basin is a very powerful filter in time and space which smooths significantly an input impulse. Some studies, mostly dealing with flash floods, have been performed to determine the characteristics of rainfall which explain the variations of hydrographs at the basin outlet and reach different conclusions. p17. 13-16: It is interesting to keep in mind that a basin is a geographic system, not only characterized by its outlet. A distributed model allows to determine the flow at any location within the basin, if the rainfall is measured at corresponding scales.

AR: Thank you for the interesting comment. We agree that this is a good point to clarify, highlighting also the distinction between runoff area and runoff model grid.

RC: p17. 18.32 (paragraph 6.1.1): this reasoning applies to calculate the flow at the basin outlet, and it is no more valid if to get the flow at locations within the basin. p17. 25-26: I would suggest be very careful with these relations, which remain only indicative, and subject to a large mean error. For instance, it is very different from the equation 2. I would suggest to keep a critical and consistent approach along the manuscript. p17-18 (paragraph 6.1.2): I suggest to regroup it with 6.1.1. both deal the influence of rainfall variability according to the basin features : surface, length, response time . . .. p17-18. The subject of these two paragraphs (to be regrouped) - influence of spatial and temporal rainfall variability in relation with basin characteristics – is very important, and has been addressed by a large number of papers. The authors present in detail a limited number of studies (five or six). I suggest that they enrich their bibliography on that subject in order to provide a more comprehensive state of the art on that subject.

[Figure]

Lanza LG, Stagi L., 2009. High resolution performance of catching type rain gauges from the laboratory phase of the WMO Field Intercomparison of Rain Intensity Gauges. Atmos Res., 94, 555-563. Lanza L.G., Vuerich E., 2009. The WMO Field Intercomparison of Rain Intensity Gauges. Atmos. Res., 94,4, 534-543.

AR: The authors agree that the number of references is limited and some other references will be added (as for example Peleg et al., 2016). However, in urban areas, not too much has been done in this field, and most of the innovative and recent works are already included in the manuscript.

Peleg, N., Blumensaat, F., Molnar, P., Fatichi, S., and Burlando, P.: Partitioning spatial and temporal rainfall variability in urban drainage modelling, Hydrol. Earth Syst. Sci. Discuss., doi:10.5194/hess-2016-530, in review, 2016.

---

## Editor Comment (EC1) · P. Molnar (Editor) · 22 Feb 2017

The reviews and author responses in the discussion of the paper "The Spatial and temporal variability of rainfall and their effects on hydrological response in urban areas – a review" are illustrative of the progress that has been made in understanding the role of space-time variability in rainfall and also of the open issues that deserve further attention.

I would specially concur with the referees that (a) the literature is addressed more comprehensively – both referees have provided suggestions on papers that may be included in your review provided they are an added-value; and (b) focus is more clearly placed on urban studies as this is the goal of the paper in the first place – in this sense

results from natural catchments are mostly interesting insofar they provide a reference that is relevant for the response of systems to space-time rainfall in general.

I find that the authors have a good plan to revise the paper and in the response letters I read clear indications of how and where the paper will be improved. We are looking forward to receiving the revised manuscript.

Peter Molnar

---

## Author Response (AR1)

**Comments: Reply to Reviewer #1**

General

The manuscript entitled "Spatial and temporal variability of rainfall and their effects on hydrological response in urban areas - a review" by Cristiano et al. provides a literature review of the current understanding of hydrological processes in urban environments with a focus on spatial and temporal variability and scales. It is well written and understandable and would fit well into the scope of HESS special issue on rainfall and urban hydrology. I found no major issues or concerns to be addressed while reading the manuscript, but I have some suggestions of expending the content discussed in part of the sections. Please find my comments, corrections and suggestions below.

Specific comments [Page; Lines]

Thank you for catching the typo mistakes. They have been corrected

[1; 19-20] For that you can also add the relatively new use of high-quality imagery from unmanned aerial vehicles (UAVs). See the paper by Tokarczyk et al. (2015).

Tokarczyk, P., Leitao, J. P., Rieckermann, J., Schindler, K., and Blumensaat, F.: High-quality observation of surface imperviousness for urban runoff modelling using UAV imagery, Hydrol. Earth Syst. Sci., 19, 4215-4228, doi:10.5194/hess-19-4215-2015, 2015.

Thank you for the suggestion. The reference has been added.

[2; 5] "many aspects" – such as?

The ambiguous sentence was rephrased as follows: "Despite these efforts, many aspects of hydrological processes in urban areas remain poorly understood, especially concerning the interaction between rainfall and runoff".

[2; 13-14] "Section 7, main knowledge gaps are identified for the with respect to accurate

prediction" – please revise.

The sentence was corrected: "In Section 7, main knowledge gaps are identified with respect to …"

[3; 3] The title is not completely accurate as you specifically refer to downscaling and upscaling of climate variable to be used as input into hydrological models.

Thank you for the suggestion. The title was changed as suggested:" Downscaling and upscaling of climate variables used as input for hydrological models"

[3; 9] "meteohydrology" – I believe that the term "hydrometeorology" is more common.

Thank you. The term "hydrometeorology" is now used

[3; 13] "Muthusamy et al., 2016" – this paper discusses upscaling rather than downscaling. I suggested changing the beginning of the sentence to "Statistical downscaling and upscaling approaches … ".

Thank you for the correction. The cited reference is indeed referring to upscaling. The sentence was rephrased as suggested.

[3; 14] In addition to the paper by Wilby and Wigley (1997) I would also suggest the authors to add the review paper of Wilks and Wilby from 1999 and the (relatively newer) paper by Fowler et al (2007). After all, this is a review paper that should cover the benchmarked papers in the field.

Fowler, H. J., Blenkinsop, S., and Tebaldi, C.: Linking climate change modelling to impacts studies: recent advances in downscaling techniques for hydrological modelling, International Journal of Climatology, 27, 1547–1578, doi:10.1002/joc.1556, 2007.

Wilks, D. S. and Wilby, R. L.: The weather generation game: a review of stochastic weather models, Progress in Physical Geography, 23, 329–357, 1999.

Thank you for the good suggestions. The references have been added.

[3; 18-19] I believe that some progress has been made in the AR methods that are being used to generate distributed rainfall since the papers of Ferraris (2003) and Schertzer and Lovejoy (2011). I would suggest the authors to modify the sentence in lines 18-19 to account for some of the relatively new publications in the field. Maybe something like: "Autoregressive methods, also refer to nowadays as "rainfall generator models", are used to generate multidimensional random fields while preserving the rainfall spatial autocorrelation (e.g. Paschalis et al., 2013; Peleg and Morin, 2014; Niemi et al., 2016)".

The three references represent the state of the art high resolution rainfall generator that are now available: STREAP (Paschalis), HiReS-WG (Peleg) and STEPS (Niemi). To that you can probably add the paper by McRobie et al. (2013) in which they extended the earlier Willems model to generate spatially distributed Gaussian rainfall cells (alternatively, this can go to the last model type you are suggesting in this paragraph).

Paschalis, A., Molnar, P., Fatichi, S., and Burlando, P.: A stochastic model for high resolution space-time precipitation simulation, Water Resources Research, 49, 8400– 8417, doi:10.1002/2013WR014437, http://dx.doi.org/10.1002/2013WR014437, 2013.

Peleg, N. and Morin, E.: Stochastic convective rain-field simulation using a high-resolution synoptically conditioned weather generator (HiReS-WG),Water Resources Research, 50, 2124–2139, doi:10.1002/2013WR014836, http://dx.doi.org/10.1002/2013WR014836, 2014.

McRobie, F. H., Wang, L.-P., Onof, C., and Kenney, S.: A spatial-temporal rainfall generator for urban drainage design, Water Science and Technology, 68, 240–249, doi:10.2166/wst.2013.241, 2013.

Niemi, T. J., Guillaume, J. H. A., Kokkonen, T., 5 Hoang, T. M. T., and Seed, A. W.: Role of spatial anisotropy in design storm generation: Experiment and interpretation, Water Resources Research, 52, 69–89, doi:10.1002/2015WR017521, http://dx.doi.org/10.1002/2015WR017521, 2016.

Thank you for the useful suggestion. The first sentence has been rephrased as suggested: "Autoregressive methods, nowadays often referred to as "rainfall generator models" are used to generate multidimensional random fields while preserving the rainfall spatial autocorrelation (Paschalis et al., 2013; Peleg and Morin, 2014; Sørup et al., 2015; Niemi et al., 2016).", and the paper by McRobie et al. (2013) was added as reference to the last model type.

[4; 8-10] There is a repetition with the previous sentence.

The sentence was indeed repetitive and has been removed

[4; 30-31] "… as, for example the approximation presented by Gericke and Smithers (2014), for which tlag = 0.6tc" – consider deleting this sentence, I don't see how this example can contribute to the reader.

We agree with the reviewer that the sentence was not adding value and removed it.

[5; 9] "… the behaviour of four Israelian catchments" – instead: "… the behaviour of four rural catchments in Israel".

Thank you for the correction. The sentence was corrected as suggested

[6; 3] You are referring to five papers as an example to "urban catchments" while not referring at all to studies on "natural watershed", although there are plenty of papers to choose from. I would suggest adding 2-3 references to benchmark papers discussing the use of weather radar in rural catchments as well. If already mentioning papers that are related to radar and urban hydrology, there is also the paper by Thorndahl et al (2016) which is part of the special issue and I think should also be mention in this review paper.

Thorndahl, S., Einfalt, T., Willems, P., Nielsen, J. E., ten Veldhuis, M.-C., Arnbjerg-Nielsen, K., Rasmussen, M. R., and Molnar, P.: Weather radar rainfall data in urban hydrology, Hydrol. Earth Syst. Sci. Discuss., doi:10.5194/hess-2016-517, in review, 2016.

Thank you for this suggestion. The suggested reference was added, but we preferred not to add references related to natural basins, in order to keep the focus on urban areas.

[6; 8-9] Consider also to add to this references the paper by Fencl et al. (2016), which is going to be published as part of this special issue.

Fencl, M., Dohnal, M., Rieckermann, J., and Bareš, V.: Gauge-Adjusted Rainfall Estimates from Commercial Microwave Links, Hydrol. Earth Syst. Sci. Discuss., doi:10.5194/hess- 2016-397, in review, 2016.

Thank you for suggesting the reference. It has been added.

[6; 20-21] "To solve the problem of spatial representation, interpolation techniques are used to obtain distributed rainfall fields…" – good. But sometimes you wish to do the opposite, go from a data obtained by a dense rain—gauge network to the areal rainfall that represents the catchment. This is the upscaling paper by Muthusamy et al. (2016) that was mentioned on [3; 13].

Thank you for the comment. The sentence was revised taking into account the suggested reference.

[6; 25-26] "A second problem is introduced by hard surfaces, that may cause water splashing into the gauges" – I thought that the recommendation of the WMO are to mount the gauges at an elevation of 1.2 m above ground. If this is the case, I don't think water splashing is an issue.

The sentence was rephrased as follows: "..water splashing into the gauges, if it is not placed at an elevation of at least 1.2m."

[7; 6-7] Consider presenting the comparison between the different band widths in a table. Maybe add price estimation for each radar type?

Thank you for the suggestion. A table summarizing the properties of radars has been added. Prices vary wildly due to many factors and would not be included.

[7; 14] Limitations of rain—gauges are not discussed in this section. Please remove "abd rain gauges" from the section title.

The title has been changed as suggested

[7; 19-20] "…and to define the uncertainty related to radar-rainfall estimation (Mandapaka et al., 2009; Overeem et al., 2009a)" – I suggest the authors to remove the reference to Overeem from this sentence (but to keep this reference when it cited next in the paragraph) and to replace it with other studies that were more focusing on rainfall radar uncertainties, such as: Ciach and Krajewski (1999), Villarini et al. (2008) and Peleg et al. (2013).

Ciach, G. J. and Krajewski, W. F.: On the estimation of radar rainfall error variance, Adv. Water Resour., 22, 585–595, doi:10.1016/s0309-1708(98)00043-8, 1999.

Villarini, G., Mandapaka, P. V., Krajewski, W. F., and Moore, R. J.: Rainfall and sampling uncertainties: A rain gauge perspective, J. Geophys. Res.-Atmos., 113, D11102, doi:10.1029/2007jd009214, 2008.

Peleg, N., Ben-Asher, M., and Morin, E.: Radar subpixel-scale rainfall variability and uncertainty: lessons learned from observations of a dense rain-gauge network, Hydrology and Earth System Sciences, 17, 2195–2208, doi:10.5194/hess-17-2195-2013, 2013.

Thank you for the good comment. The suggested references are indeed interesting for the study of rainfall radar uncertainties. They have been added as suggested.

[7; 32-33] A repetitive sentence. Consider deleting.

The sentence was deleted as suggested.

[8; 7-8] I think that an operative rainfall forecast based on weather radar has been activated in Belgium (using STEP model). Please check the following paper: Foresti, L., M. Reyniers, A. Seed, and L. Delobbe. "Development and verification of a realtime stochastic precipitation nowcasting system for urban hydrology in Belgium." Hydrology and Earth System Sciences 20 (2016): 505-527.

Thank you for suggesting this reference. It was added and the sentence was rephrased as follows:" Radar data can provide an accurate short term forecast and recent studies have presented nowcasting systems able to reduce errors in rainfall estimation (e. g. Foresti et al. 2016)."

[8; 10-12] Pollution due to urbanization also affects rainfall. Check for example the paper mention below. I would also suggest add to modify the sentence in line 11 accordingly: "Increase in heat and pollution produced by human activities …".

Givati, A., & Rosenfeld, D. (2004). Quantifying precipitation suppression due to air pollution. Journal of Applied meteorology, 43(7), 1038-1056.

Thank you for suggesting this reference. It has been added. Following the suggestion of reviewer #2 paragraph "Influence of urban areas on rainfall" has been removed and summarised at the beginning of section 3. "Urban areas affect the local hydrological system, not only by increasing the imperviousness degree of the soil, but also by changing rainfall generation and intensity patterns. Several studies show that increase in heat and pollution produced by human activities and changes in surface roughness influence rainfall and wind generation (Huff:1973, Shepherd:2002, Givati:2004, Shepherd:2006, Smith:2012, Daniels:2015, Salvadore:2015). This phenomenon is not deeply investigated in this paper, but it is an important aspect to consider."

[9; 6-23] This read to me as a separate subsection, entitled as "rainfall variability at the urban scale", or alike.

Thank you for the suggestion. The 2 sections have been separated as suggested

[9; 16-17] Not necessarily, the setup needed for deployment of a dense rain-gauge network at the urban scale that can well represent the rainfall spatial variability can be calculated using the variance reduction factor. See papers by Villarini et al. (2008) and Peleg et al. (2013) that were suggested above.

Thank you for the constructive comment. The paragraph has been revised; the following text was added taking into account the suggested references: "An alternative solution is to consider the variance reduction factor method, a numerical method to represent the uncertainty from averaging a number of rain gauges per pixel, taking into account their spatial distribution and the correlation between them. The variance reduction factor method was introduced for the first time by Rodriguez-Iturbe and Mejıa (1974) and lately applied in various studies (Krajewski et al., 2000; Villarini et al., 2008; Peleg et al., 2013)."

[9; 23] Please also have a look at the recent paper by Peleg et al., whom examined the spatial distribution of extreme rainfall intensity for the same scale and using similar methods (but with different rainfall model) as Gires et al. mentioned here. They found that the spatial distribution of extreme rainfall over small domains (1 x 1 km2) can be very high.

Peleg, N., Marra, F., Fatichi, S., Paschalis, A., Molnar, P., and Burlando, P.: Spatial variability of extreme rainfall at radar subpixel scale, Journal of Hydrology, doi:doi:10.1016/j.jhydrol.2016.05.033, 2016.

This reference has been added and discussed as follows: "Similar results are confirmed by Peleg et al. (2016), who studied the spatial variability of extreme rainfall at radar subpixel scale. Comparing a radar pixel of 1kmx1km with high resolution rainfall data, obtained by applying the stochastic rainfall generator STREAP (Peleg et al., 2013) to simulate rain fields, this study highlights that subpixel variability is high and increases with increasing of return period and with shorter duration."

[9; 27] What do you mean by common? C-band radars?

The word common was replaced with operational.

[10; 27-28] Please delete. It repeats what is already mention.

The sentence was deleted, as suggested.

[10; 29] Consider changing the title to: "Groundwater recharge and subsurface processes in urban areas". Infiltration is already discussed in the previous paragraph.

The title has been changed as suggested.

[11; 10-12] Please revise this sentence.

The sentence was rephrased as follows:" Several types of pervious pavements are used in urban areas. They can generally be separated in monolithic and modular structures. Monolithic structures consist of a combination of impermeable blocks of concrete and open joints or apertures that allow water to infiltrate. In the modular structures, gaps between two blocks are not filled with sand, as with conventional pavements,…."

[13; 20] A reference is needed here.

The requested reference was added.

[13; 21] Delete "Recent", a study from 1991 cannot be consider as recent…

The word "recent" was replaced by "some"

 [13; 30] Why the catchments need not to be placed on concrete or asphalt?

The sentence has been removed.

[15; 2-7] Please give references and examples to the two types of models used in urban studies.

The sentence has been removed as suggested by reviewer # 2, because it is considered ambiguous and not clear.

[16; 14-18] Please indicate full names for UDTM and EPA SWMM models. A reference for the SWMM model should also be added.

The full name of the 2 models and the reference for the SWMM model have been added.

[16; 2-24] It can be useful to add a table with the most common hydrodynamic models that are been used in urban studies (including full name, abbreviation, reference and the model type).

Thank you for this suggestion. A similar and exhaustive table was already presented in Salvadore et al. (2015). We decided to refer to this paper, instead of inserting a similar table in our manuscript: "A good summary of the most used urban hydrological models is presented by Salvatore et al. (2015), where a table containing 43 numerical studies highlights the model characteristics."

[17; 12] There is another relevant paper that is a part of this special issue (see below). It deals with the effect of spatially distributed rainfall on the flow total variability in an urban catchment.

Peleg, N., Blumensaat, F., Molnar, P., Fatichi, S., and Burlando, P.: Partitioning spatial and temporal rainfall variability in urban drainage modelling, Hydrol. Earth Syst. Sci. Discuss., doi:10.5194/hess-2016-530, in review, 2016.

The reference has been added with a few sentences that describe the paper: "To investigate the effects of spatial and temporal variability on urban hydrological response, Peleg et al. (2016 in review) used a stochastic rainfall generator to obtain high resolution spatially variable rainfall as input for a calibrated hydrodynamic model. They compared the contributions of climatological rainfall variability and spatial rainfall variability on peak flow variability, over a period of 30 years. They found that peak flow variability is mainly influenced by climatological rainfall, while the effects of spatial rainfall variability increase for longer return periods."

Figure 2 – Consider changing the pixel to a point for the point value (or add a point within the pixel).

Thank you for the suggestion. The figure has been changed and a point is added within the pixel.

**Comments: Reply to Reviewer #2**

General comments: This manuscript is a scientific review on the variability of rainfall in urban hydrology. Several review papers have recently been published on urban hydrology in the main journals. An additional one could be interesting if it proposes an original point of view. As far as I know, it seems to be the case for this manuscript. Nevertheless, I consider that the manuscript requires a significant revision before its publication in HESS can be considered. - The manuscript covers a wide range of topics, sometimes from a very general point of view (for instance disaggregation, but not only), and sometimes already very well documented in text books (for instance hydrological processes). It is not consistent with a review paper which should present the state of the art of research on the addressed subject. The authors are recommended to focus on the most original part of the manuscript, for which a review presents an added value. - I have in mind several papers, which address the subject of the manuscript, and which have been omitted. I recommend that the authors provide a more comprehensive and exhaustive state of the art, representative of the recent studies. In addition, the references to studies of urban basins (instead of, or in addition to, natural ones) would be welcome in a manuscript covering urban areas.

Specific comments

p1. 15-18: This sentence is questionable. Hydrologists have been working on rainfall radar measurement for a very long time, and have significantly contributed to this subject, including in urban hydrology (Einfalt et al., 2004 for a review). The cited references are recent and don't reflect this long term research effort which has known a renewed interest with the emergence of polarimetric X-band radars which allow to solve some problems met with classic low-cost radars.

Thank you for the suggestion. We reconsidered the sentences and we agree that the paragraph was not reflecting the long term research effort. We highlighted the emphasis on the long term effort and on the fact that it is not only a recent or future development. The suggested reference (Einfalt et al. 2004) has been added, as well as the recent and interesting paper by Thorndahl et al 2016. The sentence was rephrased as follows: "These developments have been applied in urban hydrology research (for a review Einfalt et al. (2004); Thorndahl et al. (2016)), where the hydrological response is….."

p2. 1-3: I am not sure that it is more complicated, I would say different.

Thank you for pointing out this unclear sentence, the word "complicated" could indeed be ambiguous. The sentence has been changed to: "detention  and control facilities, such as reservoirs, pumps and weirs, are additional elements…"

 p3.1-2 : I don't well understand this sentence

The sentence has been rephrased as follows: " Under the best scenario, process and observation scale should coincides, but this is not always the case, and transformations based on downscaling and upscaling techniques (Fig. 2) might be necessary to obtain the required observation scale."

p3. 3-21: Downscaling and upscaling in hydrology: This paragraph is a very brief and general introduction of downscaling methods. It is not very useful for the reader because the authors don't refer to the applications of these methods in urban hydrology, which is the subject of the manuscript.

The paragraph was lacking of specific references. It has been modified, adding the references suggested by reviewer #1(Paschalis et al., 2013; Peleg and Morin, 2014; and McRobie et al. (2013)) and Niemi et al., 2016).

p3. 27-30: the paper by Julien and Moglen (1990) doesn't address the particular case of urban catchments.

The authors agree with this comment. We mentioned the paper by Julien and Moglen (1990) as an example of spatial scale definition, while indicating that it was developed for natural catchments, yet there is no reason it could not be applied to urban catchments. We have revised this section and references to spatial scale definitions used in studies of urban catchments. A new paragraph was added: "In urban catchments, the concept of catchment length, defined as the squared root of the (sub)catchment or runoff area, has been used (Bruni et al. 2015, Ochoa-Rodriguez et al., 2015}. Additionally, Bruni et al. (2015) introduced the sewer length or inter-pipes sewer distance, as the ratio between the catchment area and the total length of the sewer, to characterize the spatial scale of sewer networks."

4. 6-11: I have read the paper by Gericke and Smithers (2014). Their review of the existing methods doesn't address the urban basins.

Thank you for the suggestion. We modified this paragraph, shortened the part dedicated to Gericke and Smithers (2014) and made clear that we mentioned this reference to highlight the difference with urban areas.

p4. 13-14: The production function is very different in natural areas and urban areas. Initial losses don't exceed 1 or 2 mm, and most of the impervious surfaces are directly connected to the hydrographic (or sewer) network. This statement is not valid for urban basins.

The statement was rephrased as follows: "In urban areas, where most of the surface is directly connected to the drainage system, concentration time is given by the time the rainfall needs to enter the sewer system and the time spent in the sewer."

p4. 24: The term response time is also often used; this term may be similar to the response time (Musy, 2011).

We assume the reviewer suggested to consider the response time proposed by Musy and Higy (2010). This reference has been added to the paper.

p4. 32-33: To the best of my knowledge, the paper by Morin et al. (2001) doesn't address urban basins

The work done by Morin et al. (2001) is indeed referring to natural catchments. This paragraph was shortened and the specification that it refers to natural catchments was added. We believe it is important to keep this reference as it has potential for application in urban areas.

paragraph 2.3.2: "time scale characteristics". It is interesting to highlight these terms which characterize the basin dynamics are less used nowadays (see the topic called synthetic hydrology which addresses this subject). In my opinion, there are more or less equivalent, and too many words are used to name very close concepts, which can be confusing. It could be the opportunity to propose one or two terms. I think that most of the papers that are cited (not very recent) don't concern urban basins, which is a problem concerning the subject of this manuscript. I know that similar relations have been proposed (at the same period) in urban hydrology to relate the response (or lag) time of urban basins to the characteristics of basins : surface, imperviousness, slope, roughness . . . I recommend to add references on time scale characteristics that address urban basins.

Thank you for this suggestion. Paragraph 2.3.2 was revised and references related to urban catchments were added, as follows:" In this section, we present a brief overview of time scales reported in the literature and discuss approaches to estimate characteristic time scales that have been specifically developed for urban areas. A summary of time scale characteristics if presented in Table 1. The first method to investigate the hydrological response is the rational method, presented more than a century ago by (Kuichling, 1889) for urban areas. This method was later adapted for rural areas. The rational method requires the estimation of the time of concentration in order to define the runoff volume. Time of concentration $tc$ is one of the most common hydrological characteristic time scales and it is defined as the time that a drop that falls on the most remote part of the basin needs to reach the basin outlet (Singh, 1997; Musy and Higy, 2010). Several equations to estimate this parameter are available in the literature for natural (Gericke and Smithers, 2014) and urban (McCuen et al., 1984) catchments. The time of concentration is difficult to measure, because it assumes that initial losses are already satisfied and the rainfall event intensity is constant for a period at least as long as the time of concentration. Different theoretical definitions have been developed in order to estimate the time of concentration as function of basin length, slope and other characteristics (see for some examples Singh (1976); Morin et al. (2001); USDA (2010); Gericke and Smithers (2014)).

Due to difficulties related to the estimation of time of concentration, Larson (1965) introduced the time of virtual equilibrium $t_{ve}$, defined as the time until response is 97% of runoff supply. When a given rainfall rate persists on a region for enough time to reach the equilibrium, this time is called time to equilibrium $t_e$ (Ogden et al., 1995; Ogden and Dawdy, 2003; van de Giesen et al., 2005). Time of equilibrium for a turbulent flow on a rectangular runoff plane given rainfall intensity $i$, with given roughness n, length $L_p$ and slope S can be written as (Ogden et al., 1995):

$$te = [nLp/(S^{1/2}i^{2/3})]^{3/5}$$

Another commonly used hydrological characteristic time scale or response time is the lag time $t_{lag}$. It represents the delay between rainfall and runoff generation. $t_{lag}$ is defined as the distance between the hyetograph and hydrograph center of mass of (Berne et al., 2004), or between the time of rainfall peak and time of flow peak (Marchi et al., 2010; Yao et al., 2016). $t_{lag}$ can be considered characteristic of a basin, and is dependent on drainage area, imperviousness and slope (Morin et al., 2001; Berne et al., 2004; Yao et al., 2016). Berne et al. (2004), including the results of Schaake and Knapp (1967) and Morin et al. (2001), defined a relation between the dimension of the catchment area S (in ha) and the lag time $t_{lag}$ (in mm): $t_{lag} = 3S^{0.3}$ for urban areas. Empirical relations between $t_{lag}$ and $t_c$ are presented in the literature (USDA, 2010; Gericke and Smithers, 2014).

Another characteristic time scale is the 'response time scale' $T_s$, presented for the first time by Morin et al. (2001). It is defined as the time scale at which the pattern of the time averaged and basin

averaged radar rainfall hyetograph is most similar to the pattern of the measured hydrograph at the outlet of the basin. This definition was updated by Morin et al. (2002), that used an objective and automatic algorithm to analyse the smoothness of the hyetograph and hydrograph instead of the general behaviour, and by Shamir et al. (2005), who related the number of peaks with the total duration of the rising and declining limbs of hyetographs a and hydrographs.
In urban areas, where most of the surface is directly connected to the drainage system, concentration time is given by the time the rainfall needs to enter the sewer system and the travel time through the sewer system."

p6. 15-20: I suggest to refer to Lanza and Stagi (2009) and or to Lanza and Vuerich (2009) for a recent evaluation lab. and field evaluation and comparison of rain gauges.

Thank you, the suggested references were added.

p6. 18-19: the rain gauge data is punctual, but as rainfall field displays a spatial organization, this data is representative of rainfall in its neighborhood, in a surface area which depends on time step and decorrelation distance, itself related to the rainfall type. P6. 21: It exists in the scientific literature many papers, including review papers, dealing with the interpolation of rain gauges data for mapping rainfall fields by various methods, and I don't understand why the authors refer to Shaghaghian and Abedini (2013) not published in an international journal.

The paragraph was rephrased as follows: " The main disadvantage of rain gauges is that the obtained data are point measurements and, due to the high spatial variability of rainfall events, measurements from a single rain gauges are often not representative of a larger area. Rainfall fields, however, present a spatial organization and, by interpolating data from a rain gauge networks, it is possible to obtain distributed rainfall fields (Villarini et. Al., 2008; Muthusamy et al., 2016)."

p7. 10-14: The added value of polarimetric data is mainly: i) for ground clutter detection removal, and for X-band (and C-band also) radars for attenuation correction. X-band are strongly affected by attenuation of the signal by rainfall, and the correction of this problem is very unstable. Polarimetric data allow to efficiently correct this problem.

The sentence has been rephrased as follows:" A specific strength of polarimetric radars is the use of differential phase $K_{dp}$, which allows to correct signal attenuation thus solving an important problem generally associated with X-band radars (Otto et al., 2011; Ochoa-Rodriguez et al., 2015; Thorndahl et al., 2016 in review).

p7. 15-25 : I don't agree with this paragraph for two reasons. The authors refer to a few studies dealing with the calibration of radar data by rain gauges, or the combination of radar and rain gauges data (which is a much more recent approach of this question). I think that these studies are not representative of the state of knowledge on that subject. In addition, based on a very limited number of studies, they conclude to the underestimation of rainfall by radar. I consider that this conclusion is erroneous and not justified

Thank you for the constructive comment. The paragraph was revised and several references were added, as suggested by reviewer #1 (Ciach and Krajewski, 1999; Villarini et al., 2008; Mandapaka et al., 2009; Peleg et al., 2013). Those references confirm that, generally, radar measurements underestimate rainfall when compared to rain gauges. It is however a general conclusion, and "in

most of the cases" was added in order to avoid the misunderstanding that there is always underestimation.

p7. 28: Be more precise, please! The radar equation relates the backscattered power to the radar reflectivity factor, usually called reflectivity. This radar equation doesn't depend on the drop size distribution.

Thank you for the constructive comment. The sentence was rephrased as follows: "Moreover, radar measurements need to be combined with a rain drop size distribution to obtain an accurate rainfall estimation."

p8 : paragraph 3.2 "Influence of urban areas on rainfall". This paragraph provides a brief, and general overview of this subject. What is its interaction with the subject of the manuscript? Is this paragraph useful ? I am not convinced.

We agree that this paragraph was not relevant considering the main goal of the paper. P The paragraph was reduced to a brief summary at the beginning of section 3: "Urban areas affect the local hydrological system, not only by increasing the imperviousness degree of the soil, but also by changing rainfall generation and intensity patterns. Several studies show that increase in heat and pollution produced by human activities and changes in surface roughness influence rainfall and wind generation (Huff:1973, Shepherd:2002, Givati:2004, Shepherd:2006, Smith:2012, Daniels:2015, Salvadore:2015). This phenomenon is not deeply investigated in this paper, but it is an important aspect to consider in rainfall analysis in urban areas."

p8-9. Paragraph 3.3. It seems that the weather is a very convenient device to analyze the spatial and temporal scales of rain fields for urban hydrology.

Thank you for the comment. The paragraph was rephrased as follows: "Characterizations and classifications of intense rainfall events have been proposed by various authors, combining rain gauges and radar rainfall data. In particular, weather radars are used as main tools to analyse rainfall spatial and temporal scale in urban areas."

p9 to p14: section 4 Hydrological processes. This very long paragraph summarizes the main hydrological processes and the main approaches used to represent them, in natural and urban areas as well. It regroups the basic knowledge in hydrology, addressed in text books, and not suited to a review manuscript. This section must be removed.

Thank you for the useful suggestion. This section has been thoroughly revised, textbook information was removed and the discussion now focus only on the aspects relevant for urban areas.

p15. 1-6: I don't understand these sentences. Are stochastic models used in urban hydrology? What type of models for what application? I don't see its usefulness in this manuscript. It could be confusing for the reader. Please, remove it!

We agree with the reviewer that this paragraph was confusing and we removed it from the manuscript.

p15. 11-14: As I understand this criterion, it concerns only the influence of the spatial variability of rainfall. There are many other factors involved in the choice of an hydrological model, depending on the applications of this model.

Thank you for this useful comment. The criterion proposed was only an example, and to highlight this the paragraph was rephrased as follows: "For example Berne et al. (2004) suggested a guideline for choosing between lumped and distributed modelling considering the representative surface associated to a single rain gauge $S_r$ [L^2]."

p15. 24-31. The notion of physically-based or conceptual is valid at a given scale (in my opinion). The computational power is no more a problem, but I agree that the parameterization of a model, and the values to assign to these parameters remains a key issue.

Thank you, we agree with the reviewer's comment.

 p16. 2-3 : it is an important element for all models at all scales

Thank you for the comment. The sentence "especially at small scale" was removed, to avoid the misunderstanding.

p16. 14-25: Meselhe et al. (2008) : I have not found this article in WRR. This example is certainly interesting. Unfortunately, it doesn't deal with urban hydrology. In a review manuscript addressing the hydrological response of urban basins, it is highly recommended to refer to papers which address the subject of the manuscript.

We agree that this article is not relevant in relation with the focus of the paper and removed the reference.

p17. 1-12: I would say that this subject is yet an open research subject, and the influence of spatial variability of rainfall on the basin response at its outlet is not yet well understood. A basin is a very powerful filter in time and space which smooths significantly an input impulse. Some studies, mostly dealing with flash floods, have been performed to determine the characteristics of rainfall which explain the variations of hydrographs at the basin outlet and reach different conclusions.

Thank you for this suggestion, we rephrased this to bring out this point more clearly, as follows: "These studies have shown how catchments act as filters in space and time for hydrological response to rainfall, delaying peaks and smoothing the intensity. However, the influence of spatial variability of rainfall on catchment response in urban areas is complex and remains an open research subject."

 p17. 13-16: It is interesting to keep in mind that a basin is a geographic system, not only characterized by its outlet. A distributed model allows to determine the flow at any location within the basin, if the rainfall is measured at corresponding scales.

Thank you for this comment. We added the following text to section 5.1 to include this interesting consideration: "The flow can be estimated at any location within the basin and not only at the catchment outlet. This is, however possible only if the rainfall is provided at an appropriate scale."

p17. 18.32 (paragraph 6.1.1): this reasoning applies to calculate the flow at the basin outlet, and it is no more valid if to get the flow at locations within the basin.

Agreed. The process is applicable to a specific range of basins, or sub-basins within the main basin, with a surface area that varies between 10ha and 100000ha, as indicated in the text..

p17. 25-26: I would suggest be very careful with these relations, which remain only indicative, and subject to a large mean error. For instance, it is very different from the equation 2. I would suggest to keep a critical and consistent approach along the manuscript.

Thank you for notice this error. There was a typing mistake in this formula, and for this reason the two equation were looking so different. However, the relations referred to specific study cases and are obtained fitting the empirical data.

p17-18 (paragraph 6.1.2): I suggest to regroup it with 6.1.1. both deal the influence of rainfall variability according to the basin features : surface, length, response time . . ..

Thank you, we followed this suggestion and merged the two paragraphs into section 6.2, changing the title as follows: "Influence of spatial and temporal rainfall variability in relation to catchment dimensions"

p17-18. The subject of these two paragraphs (to be regrouped) - influence of spatial and temporal rainfall variability in relation with basin characteristics – is very important, and has been addressed by a large number of papers. The authors present in detail a limited number of studies (five or six). I suggest that they enrich their bibliography on that subject in order to provide a more comprehensive state of the art on that subject.

Thank you for the interesting suggestion. We have done another search to find additional literature, and added new references (Wright et al., 2013 and Peleg et al. 2016). We also added a brief discussion on studies by Smith et al (2002, 2005), who studied relationships between rainfall scale and catchment scale, based on rainfall spatial coverage.

[revised manuscript text omitted]

---

## Author Response (AR2)

**Editor Decision:** Publish subject to minor revisions (further review by Editor) (15 Jun 2017) by Peter Molnar

Comments to the Author:

Both of the original referees have reviewed your revised manuscript and are overall satisfied with the revisions and the better focus of the review. Both referees suggest some minor corrections which I would like to ask you to implement. Referee 1 also suggests a closer look at Section 6.3 on the spatial vs temporal resolution issues, where the referee believes more clarity regarding the message is needed. I agree with this statement. Please list your responses to the main corrections suggested if you disagree with them.

It was also raised that the very short Discussion and Conclusion sections are somewhat overlapping. I do not have strong opinion on that. I will leave it up to you to reflect on what (if anything) could be improved in that regard.

In summary, I accept the paper for publication with minor revisions, which I will check personally and not send out for re-review. Thank you for your patience. Congratulations on your good work.

Best regards, Peter Molnar

**Reply to the editor decision**

We would like to thank the editor and the referees for the time and effort spend in reviewing this manuscript.

We have addressed the comments and corrections proposed by the referee. About Section 6.3, we reconsidered and reorganized the section, making a clearer distinction between Section 6.2 and 6.3, and focusing this section only on the effects of spatial and temporal variability.

Regarding Discussion and conclusion sections, we decided to merge them in one section, avoiding in this way the overlapping.

**RC:** Referee comment

**AR:** Author Reply

**RC:** Dear Authors,

The revise version of the manuscript is much clearer and cleaner than the first submitted version, but I still found some imprecision in the text. I think that the manuscript will benefit from another revision of the text. I mainly recommend you to revise Section 6.3 and consider revising the discussion and conclusions section in the manuscript (see my comments below).

**RC:** Specific comments

[Page Lines]

[3 3] "observation scale" – process scale?

**AR: The sentence was rephrased as follows, to avoid misunderstanding: "… to obtain the required match between scales"**

**RC:** [3 4] "climate variables" – should be rainfall instead as rainfall is the only climate variable discussed in this section.

**AR: The subsection title has been changed to "Rainfall downscaling"**

**RC:** [3 9-12] Please add references to support these statements.

**AR: The reference Xu et al. (1999) was added.**

**RC:** [3 14] Although their work is important, I am pretty sure that Fowler et al were not the first to use downscaling for climate change impact study.

**AR: The reference to Fowler et al. was not meant to be considered as the first use of downscaling for climate change. To avoid this possible misunderstanding this paragraph was moved to the end of the section and rephrased as follows: "The importance of using downscaling methods was discussed by Fowler et al. (2007), in a work where they investigated what can be learned."**

**RC:** [3 24] "convective cells" – rainfall cells.

**AR: Replaced with "convective rainfall cells"**

**RC:** [3 27] "Muthusamy et al., 2016" – should be Muthusamy et al., 2017. Please check and update the reference list, there are some papers there that are in status of "in review" while they are already published/accepted for publication.

**AR: This and the other references have been checked and updated, adding the full reference in case they have been already published.**

**RC:** [3 28] "weather generator" – rainfall generator.

**AR: Corrected as suggested**

**RC:** [4 8] the variable tc is mentioned twice in the paper. Here is refer to as "rainfall duration" and later is refer to as "time of concentration". This is of course not the same. Please check that all

variables are given a unique name and definition. Moreover, sometimes you indicate units to the variables and sometimes the variables are given without units, please be consists.

**AR: the duration is now called "d"**

RC: [7 10] In your discussion here (and later) I am missing the credit to Marshall–Palmer pioneer work (1948).

**AR: The cited reference was added.**

RC: [7 17] "larger more"

**AR: Corrected with "larger"**

RC: [7 29] Recent studies shown the potential use of weather radar to estimate extreme rainfall intensities (e.g. Eldardiry et al., 2015; Marra and Morin, 2015 – both published in JOH). This, however, was done for much larger scales than the urban scale (so far). Consider if you would like to add this to your discussion.

**AR: Although the cited references are interesting, they are not directly related to the urban environment, and for this reason they have not been added.**

RC: [8 12-24] Entire section 3.2 – What is the significant contribution of this section to the understanding of urban drainage studies? The way I see this section now, it only inform the reader with some examples of how to classify a rainfall event. I don't see it contribution to the paper. Consider removing or revising it significantly.

**AR: We understand that the link of this paragraph to the overall review may not be entirely clear, but its placement here is a direct consequence of the previous round of comments and replies. To provide more context, we have added the following sentence to the beginning of the paragraph: "Rainfall events are characterized by several elements, such as duration, intensity, velocity and their spatial and temporal variability, and many possible classifications are presented in the literature. Some of the most used examples of rainfall classification considering the rainfall variability, are described in this section. "**

RC: [8 26] This section must start with a definition of rainfall variability (space, time, climatological variability – there are many types, which all are later discussed).

**AR: The definition of rainfall variability was added as suggested. The following text has been added: "Rainfall events are often described and classified considering they variability in space and time. Spatial variability can be defined, following Peleg et al. (2017), as "the variability derived from having multiple spatially distributed rainfall fields for a given point in time". Peleg et al. (2017) introduced also the definition of climatological variability as the variability obtained from multiple climate trajectories that produce different storm distributions and rainfall intensities in time.**

RC: [8 29] "decorrelation distance" – not all readers will know what you mean, please define.

**AR: The definition of decorrelation distance has been added in section 3.3. The following text has been added: " These groups are defined considering the decorrelation distance (and decorrelation time), defined as distance (and time) from which two points show independent statistical behaviour, and it is obtained as the range of the climatological variogram (Emmanuel et al. 2012)."**

RC: [9 16] "Peleg et al., 2013" – this should be "Paschalis et al., 2013".

**AR: The reference was changed.**

**RC:** [10 15] "Although it is well known that not all rainfall turns into runoff" – this is a strong sentence, especially when later you give some examples where a large fraction of the rainfall is lost to ET even in urban catchments.

**AR: The sentence is indeed strong, but it is also common practice consider the losses as negligible, or to underestimate them.**

**RC:** [11 12] "land cover" – in this section you mainly focus on different imperviousness and antecedent soil moisture conditions. You immediately start with a few examples, but first you need to explain the readers what you mean by land cover (roads? different vegetation? roof-types?).

**AR: A short definition of land cover is presented as the beginning of the section 4.3 as introduction. The following text has been added: "In urban areas, the land cover, represented by an alternation of impervious surfaces, such as roads and roofs, and small pervious areas, such as gardens, vegetation and parks, shows a high variability in space."**

**RC:** [11 13-17] In a way this paragraph fit better to the previous or next section (4.2 or 4.4).

**AR: the paragraph was moved at the beginning of section 4.1, introduced by the following sentence: "During the process of transformation of rainfall in runoff, part of the water is lost due to several phenomena, such as infiltration, storage or evaporation."**

**RC:** [12 15-16] "Errors in estimation of annual evaporation in urban areas may still be higher than 20% (van de Ven, 1990)" – I think that if you made this type of a statement it needs to be supported with a more recent reference.

**AR: The text has been modified adding more recent references. The text was changed as follows:" Different techniques and approaches have been developed to measure and estimate the impact of evaporation, from the standard lysimeter to the use of remote sensing (Nuori et al. 2013), to the combined used of remote sensing and ground measurements (Hart et al. 2009). Different models to estimate evaporation in urban areas have been proposed (Marasco et al., 2015; Litvak et al., 2017). Litvak et al. (2017) estimated evaporation in the urban area of Los Angeles, as combination of empirical models of turfgrass evaporation and tree transpiration derived from in situ measurements. Evaporation from non-vegetated areas appears to be negligible compared with the vegetation, and turfgrass was responsible for 70% of evaporation from vegetated areas."**

**RC:** [13 3] "80%" – for small events (less than 6 mm of rainfall, if I got it right from Versini et al. paper) and on monthly basis. Please be more precise here, as 80% reduction in peak discharge can sound like a large figure while the absolute differences can be relatively small.

**AR: The sentence has been rephrased as follows to better clarify: "They showed that green roofs can reduce runoff generation in terms of peak discharge, depending on the rainfall event and initial conditions. The reduction can be up to 80% for small events, with an intensity lower than 6mm"**

**RC:** [13 7-9] Can you supply some references to past studies?

**AR: The reference to the most recent literature reviews (Zoppou (2001), Fletcher et al. (2013) and Salvadore et al. (2015)) has been added, with the following text: "(see Zoppou (2001) and Fletcher et al. (2013) for a review). A good summary of the most used urban hydrological models has been recently proposed by Salvadore et al. (2015), where a table with the most used hydrological models is presented and discussed."**

**RC:** [13 27] "appropriate scale" – do you mean gridded/spatially distributed?

**AR: Rephrased as follows: "only if the rainfall is provided with an appropriate spatial resolution"**

**RC:** [14 23] "…lack of representation of private connections" – this is not clear to me.

**AR: Rephrased as follows to clarify: "lack of knowledge about private pipe connections".**

**RC:** [15 25] "temporal variability" – should be climatological variability.

**AR: Corrected as suggested.**

**RC:** [15 25] "Peleg et al. (2016)" – should be Peleg et al. (2017) and the reference to needs to be add to the reference list: Peleg, N., Blumensaat, F., Molnar, P., Fatichi, S., and Burlando, P.: Partitioning the impacts of spatial and climatological rainfall variability in urban drainage modeling, Hydrol. Earth Syst. Sci., 21, 1559-1572, doi:10.5194/hess-21-1559-2017, 2017.

**AR: The reference was updated.**

**RC:** [16 4] "SST" can be removed.

**AR: Removed as suggested.**

**RC:** [17 10] "Rainfall required resolutions is higher for small basins" – please revise.

**AR: The sentence was rephrased as follows: "Resolution required to measure rainfall for small basin is usually high, as…"**

**RC:** [18 8] I think you are missing power at the right side of the equation.

**AR: Indeed. Corrected.**

**RC:** [18 27] "Authors" – which?

**AR: Replaced with "Ochoa-Rodriguez et al. (2016) investigated also…"**

**RC:** Section 6.3 – I think this section is a most for the review, however I suggest the authors to revise it thoroughly. There are a lot of numbers and variables that are defined in this section and it is not clear what do message do you want the reader to have at the end. For example: the scaling factors at [18 13], how one can use them in urban studies? It is not clear. Another example: [19 6-7] seems to refer to the difference between spatial and temporal variability of rainfall (section 6.1) and not to the difference in resolution (which this section should address).

**AR: The Authors agree that there was a bit of confusion between section 6.1 and 6.3, especially in the description of the work of Ochoa-Rodriguez et al. 2016. The paragraph has been reconsidered and revised. The following paragraph was moved to section 6.2: "They investigated the impact of 16 combinations of 4 different spatial resolutions (100x100 m, 500x500 m, 1000x1000 m, and 3000x3000 m) combined with 4 different temporal resolutions (1, 3, 5 and 10 min). Resolution combinations were chosen considering different aspects, such as the operational resolution of radar and rain gauges networks, characteristics temporal and spatial scale."**

**Section 6.3 was modified as follows:" As it was already discussed in previous sections, there is a dependency between spatial and temporal rainfall required resolution and they affect in a different way the hydrological response (Marsan et al., 1996; Singh, 1997; Berne et al., 2004; Gires et al., 10 2011; Ochoa-Rodriguez et al., 2015b).**

A first interaction between spatial and temporal rainfall scale was defined based on the assumption that atmospheric properties are valid also for rainfall. Following this assumption, Kolgomorov's theory (Kolgomorov, 1962) was combined with the scaling properties of the Navier-Stokes equation, in order to define a relation between space and time variability. For large Reynolds numbers, in fact, Navier-Stokes equation is invariant under scale transformations (Marsan et al., 1996; Deidda, 2000; Gires et al., 2011), and in this way temporal and spatial "scale changing" operator can be defined by dividing space and time (s and t) by scaling factors $\lambda_s$ and $\lambda_t$, relatively $s \rightarrow s/\lambda_s$ and $t \rightarrow t/\lambda_t$.

For scaling processes, there is a relation between scaling factors in time and space to take into account, that is represented the anisotropy coefficient $H_t$: $\lambda_t = \lambda^{(1-H_t)}_s$. $H_t$ is a priori unknown for rainfall, but it can be assumed equal to 1/3, a value that characterise atmospheric turbulence (Marsan et al., 1996; Gires et al., 2011, 2012). Lovejoy and Schertzer (1991) estimated $H_t = 0{,}5 \pm 0{,}3$ for raindrops. An example of application of this theory in a rainfall downscaling process is given by Gires et al. (2012): here, the rainfall is measured with a certain spatial resolution s and temporal resolution t. They hypothesised to downscale the radar pixels, dividing the length by a scaling factor $\Delta s = 3$, to obtain 9 pixels out of one. In this case, to keep the relation between spatial and temporal resolution, the duration of the time step has to be divided by a scaling factor: $\lambda_t = \lambda_s^{(1-1/3)} = 2^{(2/3)} \approx 2$.

Studying the hydrological response of the south-east French Mediterranean coast, Berne et al. (2004) proposed another relationship between spatial $\Delta s$ and temporal $\Delta t$ resolution used to measure rainfall, as: $\Delta s = 1{,}5 \sqrt{\Delta t}$ (see Section 6.2 for the formula derivation).

Ochoa-Rodriguez et al. (2015b) derived the theoretically required spatial rainfall resolution for urban hydrological modelling starting from a climatological variogram, that characterised average spatial structure of rainfall fields over the peak storm period, fitted with an exponential variogram model. They defined characteristic length scale $r_c$ of a storm event as $r_c = (\sqrt{2\pi}/3)r$, where r is the variogram range. The minimum required spatial resolution for adequate modelling of urban hydrological response was defined as half characteristic length scale of the storm: $\Delta s = r_c/2 \approx 0.418r$. The theoretically required temporal resolution $\Delta t$, was defined based on the time needed for a storm to move over distance equal to the characteristic length scale of the storm event $r_c$. It can be written as: $\Delta t = r_c/v$; where v is the magnitude of the mean storm velocity, obtained from average of the velocity vectors (magnitude and direction) estimated at each time step. Ochoa-Rodriguez et al. (2015b) investigated also the impact of different combinations of spatial and temporal resolutions as described in Section 6.2. One of the criteria used to choose some of the resolution combination was the already discussed in the literature (Berne et al., 2004), and according to Kolgomorov's scaling theory (Kolgomorov, 1962). Results showed that hydrodynamic models are more sensitive to the coarsening of temporal resolution of rainfall inputs than to the coarsening of spatial resolution, especially for fast moving storms.

In this work, the authors presented also a relation between spatial and temporal critical rainfall resolutions depending on drainage area (Table 4). For small catchments, with area smaller than 1 ha, was found to be equal to 100x100 m and 1 min, while for areas between 1 ha and 100 ha, a spatial resolution of 500x500 m can be sufficient to estimate the hydrological response. The critical spatial resolution found is lower than 5 min, for catchment size from about 250 to 900 ha. Results were confirmed by Yang et al. (2016), that presented an analysis of flash flooding in two small urban subcatchments of Harry's Brook (Princeton, New Jersey, US), focusing on the influence of rainfall variability of storm events on hydrological response.

Spatial variability seems to influence timing of runoff hydrograph, while temporal variability mainly influences peak value Singh (1997).

(Ochoa-Rodriguez et al., 2015b) investigated the influence of spatial and temporal scaling factor introduced at the beginning of this section, on runoff estimation from different input, introducing also a combined spatio-temporal factor $\Theta st$. This factor was defined using the anisotropy coefficient Ht as: $\Theta st = (\Delta Sr/\Delta S)(\Delta t/\Delta tr)^{\wedge}(1/(1-Ht))$, where $\Delta S$ and $\Delta tr$ are the required spatial and temporal resolutions (described in Section 6.2), $\Delta S$ and $\Delta t$ are the space and time resolutions used as input for model simulations and Ht is the scaling anisotropy factor. The stronger relation between drainage area and combined spatio-temporal factor $\Delta st$ compared to the relation with singular spatial or temporal scaling factor suggests that the effects of space and time has to be considered together. However, the combined effects of spatial and temporal resolution on the sensitivity to hydrological response requires future works and deeper investigations.

These studies highlighted the relatively more important role of temporal variability compared to spatial variability, for extreme rainfall events. The impact of the spatial variability, seemed to decrease with increase of total rainfall accumulation."

RC: Discussion and conclusions – I feel there is some mix between these sections. First, I am not sure if in a review paper a "Conclusions" section is a most. The discussion is now more of a summary of the key points of your review (and in that sense, some of the points in the conclusion section can be moved to the discussion section). I would like to see a discussion that is focused around the current gaps in knowledge – what do we don't know, where the research focus should be, etc.

AR: The authors agree that there is an overlap between the 2 mentioned section. To avoid it, we decided to combine the 2 sections in one (Summary and future directions). Here, a summary of the points discussed in the paper is presented, highlighting the gaps still presented in the literature.

The text has been replaced as follows:" In this article, the state of the art of spatial and temporal variability impacts of rainfall and catchment characteristics on hydrological response in urban areas has been presented. The main key points and conclusion of this study are the following.

A first aspect that has been highlighted is the high variability in space and time of hydrological processes and phenomena in urban environments. Measuring, understanding and effectively characterising temporal and spatial variability at small-scales is therefore of utmost importance. High resolution data are essential given the high variability of catchment characteristics and hydrological processes, such as infiltration, evaporation and surface runoff. An important role in urban areas is played by drainage infrastructures that highly affect the hydrological response, while in some cases the effects of these structures are not perfectly understood. Current methods and instruments often have insufficient capability to measure the considered process at their relevant scales.

Several definitions to classify time scale characteristics are available in the literature, such as time of concentration, lag time, time of equilibrium and response time scale. However, measurement or estimation of those parameters is often ambiguous, which implies a high level of uncertainty. Thus far, no common agreement has emerged on a unique set of parameters able to characterize small-scale variability of urban catchments in a way that enhances our understanding of urban hydrological response. Improved rainfall measurements have also allowed to investigate the relations between temporal and spatial rainfall scale. Relations have been presented, mostly adapting the Kolgomorov's theory to rainfall, to define the interaction between spatial and temporal scale in atmosphere. A unique relationship has not yet been found. This highlights the need for methods that can better characterise spatial and temporal scale parameters of rainfall and urban catchments in an effective way.

Uncertainty associated with rainfall spatial and temporal variability is one of the main sources of error in the estimation of hydrological response in urban areas. New technologies have been developed to measure rainfall spatial and temporal variability more accurately and at higher resolution. While rain gauges remain the most common used rainfall measurement instruments, weather radars are a promising example of recently developed instruments, able to estimate rainfall variability at high resolution. However, they still need to be combined with rain gauge networks in order to improve their accuracy. Rain gauges applied in urban areas present many limitations due to strong microclimatic variability, complicating identification of suitable locations for representative rainfall measurements. Polarimetric X-band radars combine high resolution and high accuracy measurement capability with the advantages of local installation thus avoiding overshooting and resolution loss with distance associated with large radar network. They constitute a promising direction for future urban hydrological research and rainfall and flood forecasting applications.

Many studies are reported in the literature using hydrological models with different characteristics and different representations of the catchment spatial variability. Different types of hydrological models have been developed in order to represent the spatial variability of catchment properties, such as land cover and imperviousness degree. Models can be classified based on their ability to represent the spatial variability of the catchment into lumped, semi-distributed and fully distributed models. These models have become more and more detailed, reaching high levels of spatial resolution. However, unless they are driven by similarly high resolution rainfall data, increasing model resolution cannot fundamentally improve understanding of hydrological processes or improve reliability of hydrological predictions. Infiltration, local storage, interception and evaporation are quite difficult to measure, especially in urban areas, because of the strong heterogeneity of urban land-use.

The impact of spatial and temporal rainfall variability on the hydrological response in urban areas and the role of drainage infrastructure and man-made control structures herein still remains poorly understood. It was found that sensitivity of hydrological response to spatial and temporal rainfall variability varies with catchment size, catchment shape, storm scale and storm velocity. So far, findings are mainly based on sensitivity studies using theoretical model scenarios. A wider range of conditions and scenarios based on observational datasets for urban hydrological basins need to be analysed in order to characterize better the hydrological response and its sensitivity to different spatial and temporal rainfall resolutions."

RC: Figure 2. Downscaling and upscaling arrows should be reversed. When you move from averaged area to spatially distributed points you are downscaling your data. Moreover, the arrows should not cover the range between point value (upper panel) to the pattern values (middle panel), as this process is neither downscaling nor upscaling.

AR: The figure was changed as suggested.

RC: Table 3. "intense peak inside" and "intensity rainfall lower" – not clear.

AR: Rephrased as follows: "High intensity core, combined with low intensity areas"

**Reply Referee #2**

**RC:** Referee comment

**AR:** Author Reply

**RC:** Review comments

Spatial and temporal variability of rainfall and their effects on hydrological response in urban areas - a review

Revised version by Elena Cristiano, Marie-Claire ten Veldhuis, and Nick van de Giesen

I have read the revised version of this manuscript that I reviewed in December 2016.

The manuscript have been significantly improved on various points :

- Section 2 – "Scales in Urban Hydrology" is addressed in a more comprehensive way, and better related to issues in urban hydrology. The paragraph on downscaling is more detailed. The paragraph on "time scale characteristics" is now interesting, and provides a real synthesis on that subject.

- Section 3 – "Rainfall measurement and variability in urban regions" has been revised, and I haven't comments.

- Section 5 – "Urban hydrological models" is useful, because these models are analyze the effects of rainfall variabilities

- Section 6 – "Interaction of spatial and temporal rainfall variability with hydrological in urban areas" has been revised and seems to me well addressed by the authors. It is still a recent and active research subject, with various points of view.

The bibliography should be carefully checked. I have noticed several mistakes among the papers that I know (below), and it is likely that this short list is not exhaustive.

p 26 – lines 19-20 : Berne A., Delrieu G., Creutin J.D. and C. Obled

p 26 – line 29 : the journal, issue, page numbers are missing

p 31 – line 8 : Ramier D., Berthier E. and H. Andrieu

p 31 – line 38 : Sheperd J.M., Pierce H., and A.J. Negri

**AR: The suggested mistakes and other typos have been corrected.**

[revised manuscript text omitted]